# Dispersive determination of nucleon gravitational form factors

Xiong-Hui Cao [1], Feng-Kun Guo [1,2,3,4] ✉, Qu-Zhi Li [5] & De-Liang Yao [6,7] ✉

Being closely connected to the origin of the nucleon mass, the gravitational form factors of the nucleon have attracted significant attention in recent years. We present the first model-independent determinations of the gravitational form factors of the pion and nucleon at the physical pion mass, using a data-driven dispersive approach. The so-called "last global unknown property" of the nucleon, the $D$-term, is determined to be $-3.38^{+0.34}_{-0.35}$. The root mean square radius of the scalar trace density inside the nucleon is determined to be $(0.97 \pm 0.03)$fm. Notably, this value is larger than the proton charge radius, suggesting a modern structural view of the nucleon where gluons, responsible for most of the nucleon mass, are distributed over a larger spatial region than quarks, which dominate the charge distribution, indicating that the radius of the trace density may be regarded as a confinement radius. We also predict the nucleon angular momentum and mechanical radii, providing further insights into the intricate internal structure of the nucleon.

Nucleons are fundamental building blocks of visible matter in the universe and represent the most stable bound states governed by strong interaction described by quantum chromodynamics (QCD). Through decades of dedicated work, the experimental accessibility of charge probes, which was used to show that the proton is not a point-like particle[1], has led to remarkably precise measurements of the nucleon charge radii, as reviewed in ref. 2. However, due to the extreme weakness of gravitational interaction compared to the electromagnetic one, directly obtaining the nucleon mass radius from experiments poses a significant challenge.

The gravitational structure of the hadron, encapsulated by its gravitational form factors (GFFs), is defined from the hadronic matrix elements of the QCD energy-momentum tensor (EMT). These GFFs are essential for understanding the nucleon mass, energy, angular momentum, internal stress, and other intrinsic properties[3,4]. In addition, the hadronic EMT matrix elements serve as crucial input quantities not only for the theoretical description of hadrons in gravitational fields but also for hadronic decays of heavy quarkonia[5-7],

semileptonic $\tau$ decays[8], hard exclusive processes[9-14], and even the investigations of hidden-charm pentaquarks[15,16].

The total GFFs of nucleon are defined as[17-20]

$$\left\langle N(p') \left| \hat{T}^{\mu\nu} \right| N(p) \right\rangle = \frac{1}{4m_N} \bar{u}(p') \Big[ A(t) P^\mu P^\nu + J(t) \left( iP^{\{\mu} \sigma^{\nu\}\rho} \Delta_\rho \right) + D(t) \left( \Delta^\mu \Delta^\nu - g^{\mu\nu} \Delta^2 \right) \Big] u(p),$$
(1)

where $\hat{T}^{\mu\nu}$ is the Belinfante-improved symmetric[21] and renormalization-scale-independent[22] total EMT of QCD, $a_{\{\mu} b_{\nu\}} \equiv a_\mu b_\nu + a_\nu b_\mu$, $P \equiv p' + p$, $\Delta \equiv p' - p$, $t \equiv \Delta^2$ and $\sigma_{\mu\nu} \equiv \frac{i}{2} \left[ \gamma_\mu, \gamma_\nu \right]$. The nucleon trace GFF is given as a linear combination of the above three as

$$\Theta(t) = m_N \left[ A(t) - \frac{t}{4m_N^2} (A(t) - 2J(t) + 3D(t)) \right].$$
(2)

[1]Institute of Theoretical Physics, Chinese Academy of Sciences, Beijing, China. [2]School of Physical Sciences, University of Chinese Academy of Sciences, Beijing, China. [3]Peng Huanwu Collaborative Center for Research and Education, Beihang University, Beijing, China. [4]Southern Center for Nuclear-Science Theory (SCNT), Institute of Modern Physics, Chinese Academy of Sciences, Huizhou, China. [5]Institute for Particle and Nuclear Physics, College of Physics, Sichuan University, Chengdu, Sichuan, China. [6]School of Physics and Electronics, Hunan University, Changsha, China. [7]Hunan Provincial Key Laboratory of High-Energy Scale Physics and Applications, Hunan University, Changsha, China. ✉e-mail: fkguo@itp.ac.cn; yaodeliang@hnu.edu.cn

There have been many model studies on nucleon GFFs; see, e.g., refs. [23–37]. Poincaré symmetry and on-shellness of external hadrons provide constraints in the form of GFF normalizations, $A(0) = 1$[18] and $J(0) = 1/2$[19] for the nucleon, as rigorously proven in ref. [38]. However, the D-term (Druck-term) $D \equiv D(0)$ is unconstrained by general principles in contrast to the well-known electric charge, magnetic moment, mass and spin of nucleons. It is known as the "last global unknown property" of the nucleon[3,39]. The chiral soliton models[40–45] predict a relatively broad range for the D-term, specifically $-4 \lesssim D \lesssim -1$, although they are unable to provide reliable error estimates. In principle, baryon chiral perturbation theory (ChPT)[46–50] provides the chiral representation of the GFFs systematically. However, the D-term is related to the unknown low-energy constant $c_8$[48] and cannot be predicted.

In 2021, Kharzeev[51] proposed that the mass radius of the proton from the scalar trace density could be extracted from $J/\psi$ photoproduction[52] and the fit result was ~ 0.55 fm. In fact, considerable debates persist on the validity of this connection[53,54]. Recently, calculations of nucleon GFFs from lattice QCD (LQCD) at unphysical pion masses of 170 MeV[55] and 253–539 MeV[56] became available. These LQCD calculations predicted a much larger radius ~ 1fm with uncertainties at the 10% level. Hence, a precise model-independent calculation at the physical pion mass holds paramount importance. The current work is devoted to accomplishing this task.

The theoretical toolkit is provided by dispersion relations (for a recent review, see ref. [57]). We start with the pion GFFs by considering the $\pi\pi$ and $K\bar{K}$ intermediate states and the corresponding unitarity relations. These are complemented with next-to-leading order (NLO) ChPT predictions for the normalizations and slopes of meson GFFs. By incorporating constraints from analyticity, unitarity, and sum rules, we provide a comprehensive description of the nucleon GFFs. Valuable insights into the internal static spatial distribution of nucleons then follow. Our results provide a solid foundation for future studies of the nucleon structure, and have the potential to offer new insights into strongly interacting matter at low temperatures and high baryon densities, e.g., in neutron stars[58].

## Results and discussion
### Meson form factors
Pion has two GFFs which are defined as[18,20,59–61]

$$\left\langle \pi^a(p') | \hat{T}^{\mu\nu} | \pi^b(p) \right\rangle$$
$$= \frac{\delta^{ab}}{2} \left[ A^\pi(t) P^\mu P^\nu + D^\pi(t) \left( \Delta^\mu \Delta^\nu - g^{\mu\nu} \Delta^2 \right) \right], \quad (3)$$

where $a, b = 1, 2, 3$ are isospin labels. We work in the isospin limit. Elastic unitarity gives the imaginary part from $\pi\pi$ intermediate states via the Cutkosky cutting rule[62] (see Fig. 1),

$$\text{Im}\, A^\pi(t) = \sigma_\pi(t) \left(t_2^0(t)\right)^* A^\pi(t), \quad (4)$$

$$\text{Im}\, D^\pi(t) = \sigma_\pi(t) \left[ \frac{1}{3} \sigma_\pi^2(t) \left(t_0^0(t) - t_2^0(t)\right)^* A^\pi(t) \right.$$
$$\left. + \left(t_0^0(t)\right)^* D^\pi(t) \right], \quad (5)$$

where $\sigma_i(t) \equiv \sqrt{1 - 4m_i^2/t}$ ($i = \pi, K$ and $N$) and $t_0^0(t)$ ($t_2^0(t)$) are the $S$-($D$-) wave $\pi\pi$ partial-wave amplitudes related to the phase shifts $\delta_\ell^0(t)$ according to $t_\ell^0(t) = e^{i\delta_\ell^0(t)} \sin \delta_\ell^0(t)/\sigma_\pi(t)$. Details of the derivation of Eqs. (4) and (5) are given in the Supplementary Material. In practice, the phase of the $\pi\pi$ D-wave scattering amplitude $\phi_2^0(t)$ instead of $\delta_2^0(t)$ is used to include inelastic effects. The D-wave data are taken from the latest crossing-symmetric dispersive analysis[63] instead of ref. [64] used in ref. [65]. The main difference lies in the fact that the phase shift and inelasticity from ref. [63] are consistent with the commonly used results[64] below 1.4 GeV and cover a larger energy range up to around 2 GeV. The difference turns out to be moderate.

One sees from Eq. (4) that the phase of the GFF $A^\pi$ equals $\delta_2^0$ (or $\phi_2^0$, modulo multiple of $\pi$). The dispersion relation admits a solution known as the Omnès representation[66]:

$$A^\pi(t) = (1 + \alpha t) \Omega_2^0(t), \quad (6)$$

$$\Omega_2^0(t) \equiv \exp \left\{ \frac{t}{\pi} \int_{4m_\pi^2}^\infty \frac{dt'}{t'} \frac{\phi_2^0(t')}{t' - t} \right\}. \quad (7)$$

The coefficient $\alpha$ can be estimated using the NLO ChPT result with a tensor meson dominance estimate for the relevant low-energy constant (LEC) $L_{12}^r$[59]. Namely, $\alpha = -2L_{12}^r/F_\pi^2 - \dot{\Omega}_2^0(0)$ and $L_{12}^r = -F_\pi^2/(2m_{f_2}^2)$, where $F_\pi = 92.1$MeV is the physical pion decay constant, $m_{f_2} = (1275 \pm 20)$ MeV is the mass of the $f_2(1270)$ resonance, with the uncertainty covering various experimental measurements[67] for a conservative estimate, and the dot notation indicates the derivative with respect to $t$.

However, Eq. (5) is notably more complicated because the GFF $D^\pi$ mixes the $J^{PC} = 0^{++}$ and $2^{++}$ quantum numbers, where $J$ is angular momentum (AM) and $P$, $C$ are parity and charge conjugation, respectively. We can define the pion trace GFF[68,69] $\Theta^\pi(t) = -t \left[ \sigma_\pi^2(t) A^\pi(t) + 3D^\pi(t) \right]/2$. Then Eq. (5) leads to a standard single-channel partial-wave unitarity relation $\text{Im}\, \Theta^\pi(t) = \sigma_\pi(t) \left(t_0^0(t)\right)^* \Theta^\pi(t)$, in analogy to Eq. (4).

To account for the strong $\pi\pi$-$K\bar{K}$ interactions in the $0^{++}$ channel due to the $f_0(980)$ resonance, we consider the coupled-channel Muskhelishvili-Omnès problem[66,70], given as[71]

$$\text{Im}\, \boldsymbol{\Theta}(t) = \left[\mathbf{T}_0^0(t)\right]^* \boldsymbol{\Sigma}_0^0(t) \boldsymbol{\Theta}(t), \quad (8)$$

where $\boldsymbol{\Theta}(t) = \left( \Theta^\pi(t), 2\Theta^K(t)/\sqrt{3} \right)^T$, and the definitions of T-matrix $\mathbf{T}_0^0(t)$ and phase-space factor $\boldsymbol{\Sigma}_0^0(t)$ can be found in refs. [71,72] (see also Supplementary Eqs. (27) and (29)). Using Eq. (8), the trace FFs can be

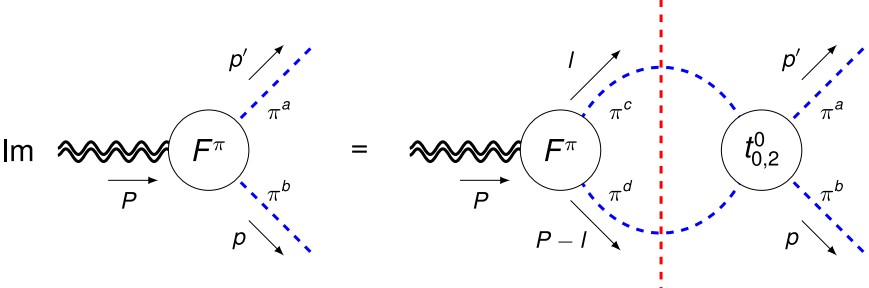

**Fig. 1 | Elastic unitarity relation for the pion GFFs $F^\pi = \{A^\pi, D^\pi\}$.** The blue dashed lines denote pions, the double wiggly lines represent the external QCD EMT current, and the red vertical dashed line indicates that the intermediate pion pair are to be taken on-shell.

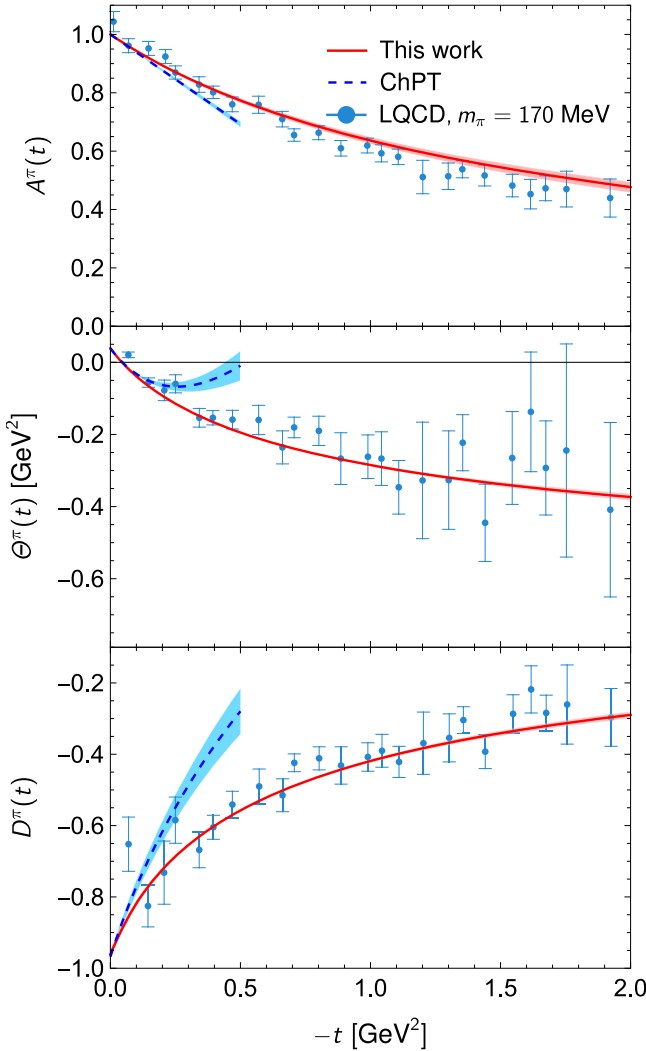

**Fig. 2 | The total GFFs $A^\pi$, $\Theta^\pi$ and $D^\pi$ of the pion.** Our predictions are shown as red solid lines. The blue dashed lines show the NLO ChPT prediction for the low $-t$ region[59]. We also show the LQCD results at $m_\pi = 170$ MeV for $A^\pi$ and $D^\pi$ in ref. 75; $\Theta^\pi$ is obtained from a linear combination of $A^\pi$ and $D^\pi$, with errors added in quadrature.

written as[71]

$$\begin{pmatrix} \Theta^\pi(t) \\ \frac{2}{\sqrt{3}} \Theta^K(t) \end{pmatrix}^T = \begin{pmatrix} 2m_\pi^2 + \beta_\pi t \\ \frac{2}{\sqrt{3}} (2m_K^2 + \beta_K t) \end{pmatrix}^T \boldsymbol{\Omega}_0^0(t), \qquad (9)$$

by virtue of the $S$-wave Omnès matrix $\boldsymbol{\Omega}_0^{0}$[72]. Notice that the parameters $\beta_\pi$ and $\beta_K$ cannot be zero due to chiral symmetry[71], and their values are related to the slopes of GFFs at $t = 0$, i.e., $\dot{\Theta}^\pi(0) = 0.98(2)$, $\dot{\Theta}^K(0) = 0.94(14)$, matching to the prediction of ChPT at NLO[59]. The uncertainties from higher order chiral corrections are much smaller than the above quoted errors and thus negligible. We refer to the Supplementary Material for further details.

We use precise phase shifts and inelasticities from analyses in refs. 63, 73, 74 as inputs. The predictions for the pion GFFs are shown in Fig. 2, where the uncertainties are obtained from the variations in $m_{f_2}$ and the slopes $\dot{\Theta}^\pi(0)$ and $\dot{\Theta}^K(0)$ mentioned above (prediction of the kaon trace GFF $\Theta^K$ is shown in Supplementary Fig. 4). We have checked that errors caused by those of the $D$-wave phase and the $S$-wave Omnès matrix are negligible. That is, the uncertainties are from the low-energy inputs from matching the dispersion representation of the meson GFFs to the NLO ChPT expressions, and can be further

reduced once the involved LECs are precisely determined from lattice QCD calculations. Our results agree well with LQCD calculations at an unphysical pion mass of 170 MeV[75].

We note that the study of pion GFFs using the dispersion approach was pioneered in ref. 71 with low-precision data, and further developed for $\Theta^\pi$ recently in ref. 69 by incorporating $S$-wave $\pi\pi$-$K\bar{K}$ scattering from dispersive analysis in ref. 8 and fitting lattice data[75]. We advance the dispersive analysis in both $\Theta^\pi$ and $A^\pi$ GFFs by utilizing precise phase shifts[63,73] and NLO ChPT results[59], achieving theoretical predictions without the need for lattice data fitting.

## Nucleon form factors

The above dispersive treatment can be generalized to the nucleon case, for which the unitarity relation is depicted in Fig. 3. Following the notation of refs. 76–78, we have

$$\text{Im}\,A(t) = \frac{3t^2\sigma_\pi^5}{32\sqrt{6}} \left[ f_-^2(t) + \frac{2\sqrt{6}m_N}{t\sigma_N^2} \Gamma^2(t) \right]^* A^\pi(t), \qquad (10)$$

$$\text{Im}\,J(t) = \frac{3t^2\sigma_\pi^5}{64\sqrt{6}} \left( f_-^2(t) \right)^* A^\pi(t), \qquad (11)$$

$$\text{Im}\,D(t) = -\frac{3m_N\sigma_\pi}{t\sigma_N^2} \left[ \frac{\sigma_\pi^2}{3} \left( f_+^0(t) - \left( \frac{t\sigma_\pi\sigma_N}{4} \right)^2 f_+^2(t) \right)^* \right.$$
$$\left. \times A^\pi(t) + \left( f_+^0(t) \right)^* D^\pi(t) \right], \qquad (12)$$

where $f_+^0(t)$ and $f_\pm^2(t)$ are the $S$- and $D$-wave amplitudes for $\pi\pi \to N\bar{N}$, and $\Gamma^2(t) \equiv m_N\sqrt{2}f_-^2(t)/\sqrt{3} - f_+^2(t)$. A detailed derivation of these equations can be found in the Supplementary Material.

Using Eqs. (10), (11) and Eq. (12), the explicit formula of the spectral function $\text{Im}\Theta$ can be written as[79]

$$\text{Im}\,\Theta(t) = -\frac{3\sigma_\pi}{2t\sigma_N^2} \left( f_+^0(t) \right)^* \Theta^\pi(t). \qquad (13)$$

It can also be generalized to include $K\bar{K}$ intermediate states,

$$\text{Im}\,\Theta(t) = -\frac{3}{2t\sigma_N^2} \left[ \sigma_\pi \left( f_+^0(t) \right)^* \Theta^\pi(t)\theta(t - 4m_\pi^2) \right.$$
$$\left. + \frac{4}{3}\sigma_K \left( h_+^0(t) \right)^* \Theta^K(t)\theta(t - 4m_K^2) \right], \qquad (14)$$

where $h_+^0$ is the $S$-wave amplitude for $K\bar{K} \to N\bar{N}$. The channel $K\bar{K}$ is important because the scalar resonance $f_0(980)$ strongly couples to $K\bar{K}$ and also to $\pi\pi$.

Once the spectral functions of nucleon GFFs are obtained from the Omnès representation of the meson GFFs in Eqs. (6), (9) and the $\pi\pi/K\bar{K} \to N\bar{N}$ partial wave amplitudes, the nucleon GFFs can be constructed from the spectral functions, by the unsubtracted dispersion relations (DRs),

$$(A, J, \Theta)(t) = \frac{1}{\pi} \int_{4m_\pi^2}^\infty dt' \frac{\text{Im}\,(A, J, \Theta)(t')}{t' - t}, \qquad (15)$$

whose convergence is ensured by the leading order perturbative QCD analyses[80–82].

One immediately obtains sum rules for the normalization of the nucleon GFFs,

$$\frac{1}{\pi} \int_{4m_\pi^2}^\infty dt' \frac{\text{Im}\,(A, J, \Theta)(t')}{t'} = \left( 1, \frac{1}{2}, m_N \right), \qquad (16)$$

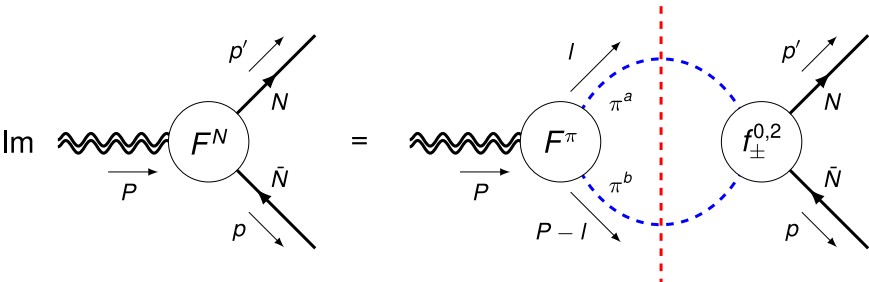

**Fig. 3 | Elastic unitarity relation for the isoscalar nucleon GFFs $F^N = \{A, J, D\}$.** The blue dashed, black solid, and double-wiggly lines denote pions, nucleons, and the external QCD EMT current, respectively; the red dashed vertical line indicates that the intermediate state $\pi\pi$ are to be taken on-shell.

and, using Eq. (2), $D$-term satisfies the following sum rule:

$$D(0) = \frac{4m_N}{3\pi} \int_{4m_\pi^2}^{\infty} dt' \frac{\text{Im}\left(m_N A(t') - \Theta(t')\right)}{t'^2} . \tag{17}$$

The sum rules in Eq. (16) serve as a strong constraint so that any violation implies breaking of the Poincaré symmetry. In fact, if the spectral functions are rigorously known, these sum rules will be satisfied. However, the integrals on the left-hand side of Eq. (16) do not always converge sufficiently fast to fully satisfy the sum rules, as also found before in the dispersive analysis of the nucleon electromagnetic form factors. To address this, following refs. 83–86, we introduce additional effective zero-width poles with masses $m_{S,D}$ into the spectral functions (10), (11) ($D$-wave) and (14) ($S$-wave), represented as $\pi c_{S,D} m_{S,D}^2 \delta\left(t - m_{S,D}^2\right)$, to simulate contributions from highly excited meson resonances. One effective pole is introduced for each partial wave, and the $S$- and $D$-wave couplings $c_{S,D}$ are fixed to ensure the sum rules (16) align with their expected values. The poles correspond to the highly excited meson resonances above ~1.4 GeV (up to about this energy the phases are precisely known) contributing to the spectral function. Their contributions are minor in the low $|t|$ region for the GFFs, and we vary the pole locations to estimate the high energy uncertainty.

Equations (10), (11), (14), and (15) are the master formulae used to compute the nucleon GFFs. The input $\pi\pi/K\bar{K} \to N\bar{N}$ $S$-wave amplitudes are from the rigorous Roy-Steiner equation analyses[65,72,79,87,88]. In particular, we take the ones from ref. 88 (see Supplementary Fig. 6). This method imposes general constraints on $\pi N$ scattering amplitudes, such as analyticity, unitarity, and crossing symmetry. The partial waves for $\pi\pi \to N\bar{N}$ are incorporated into a fully crossing-symmetric dispersive analysis, ensuring that the spectral function complies with all analytic $S$-matrix theory requirements and low-energy data constraints. The $\pi\pi$-$K\bar{K}$ two-channel approximation works very well up to about 1.3 GeV, beyond which inelasticities due to the $4\pi$ channels start to play a role. It is important to note that two subtractions were implemented in the $\pi N$ Roy-Steiner equation analysis, which significantly suppress contributions from the high-energy region[72]. The remaining high-energy contributions are accounted for by the aforementioned effective poles. This approach has been successfully applied to nucleon scalar form factors[72], the $\pi N$ $\sigma$-term[89–91], electromagnetic form factors[84,86], and antisymmetric tensor form factors[92]. For the $D$-wave contributions in Eq. (10) and Eq. (11), we adopt the results from ref. 63, which differ slightly from those in ref. 65, as noted above.

The uncertainties of our results come from three sources: (i) uncertainties of the LECs in NLO ChPT[59], which are obtained by varying $\alpha \in [-0.03, 0.01]$ GeV$^{-2}$, $\beta_\pi \in [0.68, 0.72]$ and $\beta_K \in [0.32, 0.60]$, corresponding to varying $L_{12}^r$, $\dot{\Theta}^\pi(0)$ and $\dot{\Theta}^K(0)$ as given above in the mesonic sector; (ii) uncertainties of the $\pi\pi/K\bar{K} \to N\bar{N}$ partial wave

amplitudes, which have been fully estimated in the comprehensive review of the $\pi N$ Roy-Steiner equation analysis[65]; (iii) uncertainties of the high-energy tail of the spectral functions, estimated by varying the effective pole masses. In practice, for the $S$-wave, we use one effective pole located at 1.5 ~ 1.8 GeV with the central value 1.6 GeV to cover both the $f_0(1500)$ and $f_0(1710)$ resonances; for the $D$-wave, we use one effective pole located at 1.5 ~ 2.2 GeV with the central value 1.8 GeV to cover the $f_2'(1525)$, $f_2(1565)$, $f_2(1950)$ and $f_2(2010)$ resonances. The above error budget is summarized in Table 1, where the three different sources of uncertainties are denoted as "ChPT", "pwa" and "eff", respectively.

Nevertheless, parts of the uncertainties can be further reduced in the future. For instance, the uncertainties associated with the NLO ChPT parameters can be reduced once precise LQCD data on slopes of the pion and kaon GFFs at zero momentum transfer are available; the $\pi\pi$ scattering phase shifts up to 1.8 GeV from the very recent analysis in ref. 93 can be used to improve the $\pi\pi$-$K\bar{K}$ dispersive treatment beyond ~1.4 GeV.

Our results are presented in Fig. 4. Consequently, the nucleon $D$-term is determined to be

$$D = -3.38^{+0.34}_{-0.35}, \tag{18}$$

marking the first rigorous, model-independent determination of the nucleon $D$-term at the physical pion mass. The error budget is given in Table 1. This result satisfies the positivity bound[49], $D \leq -0.20(2)$. A comparison of our result with predictions from LQCD and various models is provided in Fig. 5. It is noted that ref. 23 offers a dispersive

**Table 1 | Error budget for the $D$-term and radii for the corresponding nucleon GFFs**

| | | | | |
|---|---|---|---|---|
| $D$-term | $-3.38^{+0.34}_{-0.35}$ | $+ (0.18)_{\text{ChPT}}(0.12)_{\text{pwa}}(0.26)_{\text{eff}}$ | | |
| | | $- (0.16)_{\text{ChPT}}(0.12)_{\text{pwa}}(0.29)_{\text{eff}}$ | | |
| $\sqrt{\langle r_\Theta^2 \rangle}$ [fm] | $0.97^{+0.03}_{-0.03}$ | $+ (0.01)_{\text{ChPT}}(0.01)_{\text{pwa}}(0.03)_{\text{eff}}$ | | |
| | | $- (0.02)_{\text{ChPT}}(0.01)_{\text{pwa}}(0.02)_{\text{eff}}$ | | |
| $\sqrt{\langle r_{\text{Mass}}^2 \rangle}$ [fm] | $0.70^{+0.03}_{-0.04}$ | $+ (0.02)_{\text{ChPT}}(0.01)_{\text{pwa}}(0.02)_{\text{eff}}$ | | |
| | | $- (0.02)_{\text{ChPT}}(0.01)_{\text{pwa}}(0.03)_{\text{eff}}$ | | |
| $\sqrt{\langle r_{\text{Mech}}^2 \rangle}$ [fm] | $0.72^{+0.09}_{-0.08}$ | $+ (0.02)_{\text{ChPT}}(0.00)_{\text{pwa}}(0.09)_{\text{eff}}$ | | |
| | | $- (0.03)_{\text{ChPT}}(0.01)_{\text{pwa}}(0.07)_{\text{eff}}$ | | |
| $\sqrt{\langle r_J^2 \rangle}$ [fm] | $0.70^{+0.02}_{-0.02}$ | $+ (0.01)_{\text{ChPT}}(0.01)_{\text{pwa}}(0.01)_{\text{eff}}$ | | |
| | | $- (0.01)_{\text{ChPT}}(0.00)_{\text{pwa}}(0.02)_{\text{eff}}$ | | |

Errors in the second column are obtained by adding those in the third column in quadrature. Here "ChPT", "pwa", and "eff" refer to the errors from the NLO ChPT inputs, the partial-wave amplitudes, and the high-energy effective poles, respectively.

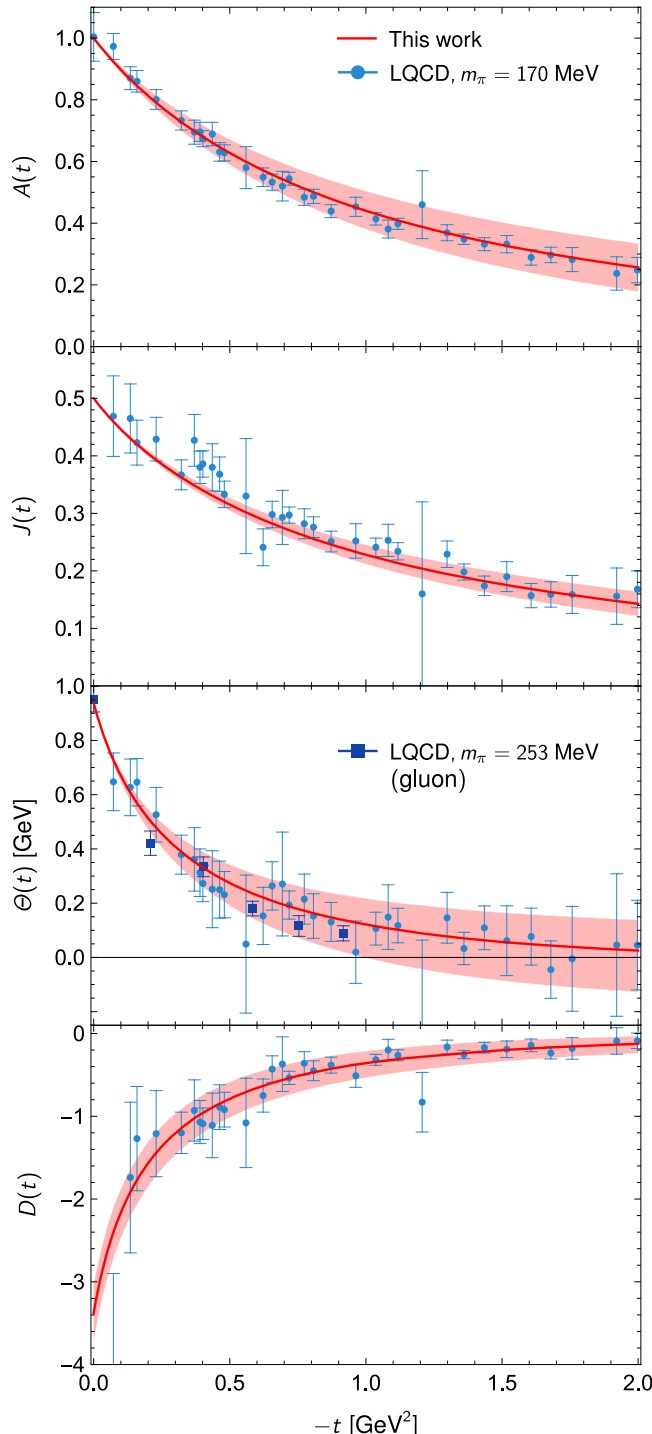

**Fig. 4 | The four total GFFs of the nucleon.** Our predictions are shown as red solid lines. We also show the LQCD results at $m_\pi = 170\,\text{MeV}$[55] and $m_\pi = 253\,\text{MeV}$[56], where the later is purely gluonic. The lattice results of $\Theta(t)$ at 170 MeV are obtained from a linear combination of the other three GFFs in ref. 55, with errors added in quadrature.

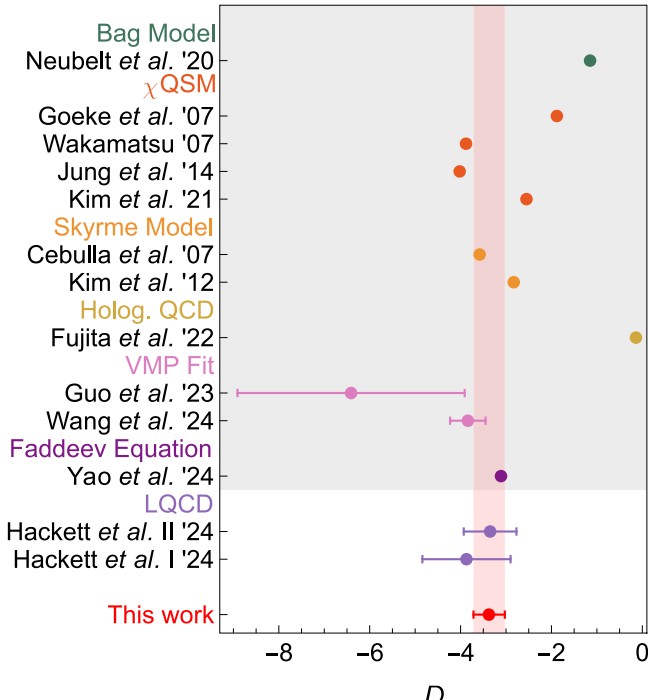

**Fig. 5 | Comparison of our result for the nucleon $D$-term with LQCD predictions**[55]. For lattice results, "I" and "II" correspond to extractions therein using tripole and $z$-expansion fits, respectively. The shaded region includes various model calculations, including Faddeev equation with the rainbow-ladder truncation[35], model fits to vector-meson ($J/\psi$) photoproduction (VMP) data[31,32], holographic QCD[106], Skyrme model[40,42], chiral quark soliton model ($\chi$QSM)[41,43-45] and bag model[25].

from the modern $\pi N$ Roy-Steiner equation analysis instead of old Karlsruhe-Helsinki results[76], and offering a reasonable estimate of uncertainties.

### Nucleon radii

Traditional chiral symmetry inspired models describe the proton as a composite system characterized by two scales[94]: a compact hard core within about 0.5 fm[51,95] and a surrounding quark-antiquark cloud (or pion cloud) in which pions play a prominent role. The core carries most of the nucleon mass generated by the (gluonic) trace anomaly, and the pion cloud surrounding this core carries the quantum numbers of the currents, giving rise to the respective form factors. Another influential picture is that a baryon can be viewed as resembling a Y-shaped string, formed by nonperturbative gluon configuration, with valence quarks at the ends[96].

Our results strongly suggest that the nucleon should be pictured differently. The mean square radius in the Breit frame of the trace GFF, i.e., derived from the matrix element of $T^\mu_\mu$[51], is determined to be

$$\langle r^2_\Theta \rangle = \frac{6\dot{\Theta}(0)}{m_N} = 6\dot{A}(0) - \frac{9D}{2m^2_N} = (0.97 \pm 0.03 \text{ fm})^2. \quad (19)$$

The mass radius, derived from the matrix element of $T^{003}$, is

$$\langle r^2_{\text{Mass}} \rangle = 6\dot{A}(0) - \frac{3D}{2m^2_N} = \left(0.70^{+0.03}_{-0.04} \text{ fm}\right)^2. \quad (20)$$

There are different definitions of the "mass radius" in the literature. In ref. 51, it is given by the radius derived from the scalar trace density, corresponding to $r_\Theta$ here. However, the term "mass radius" in ref. 97

analysis for the quark $D$-term GFF of the nucleon in deeply virtual Compton scattering. This pioneering work is limited in several aspects: model-dependent estimates of the $2\pi$ generalized distribution amplitudes, neglect of the $K\bar{K}$ intermediate states, and the absence of an error analysis. These limitations have been overcome in our work, which offers the first dispersive determination of all nucleon GFFs, by incorporating $S$-wave $\pi\pi$-$K\bar{K}$ coupled channels, using the partial waves

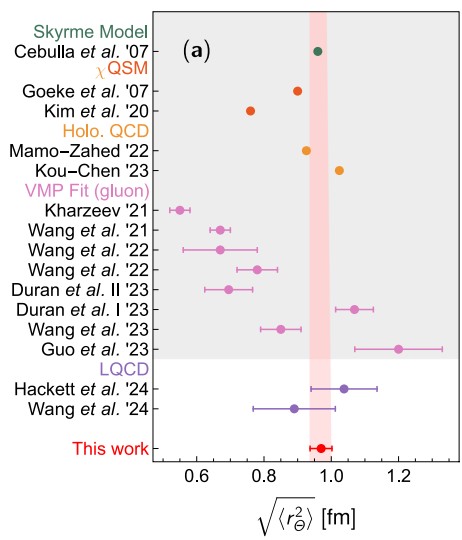
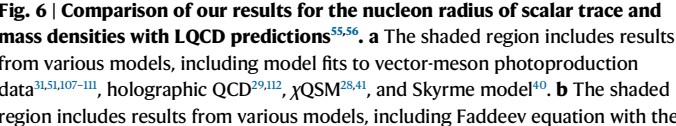
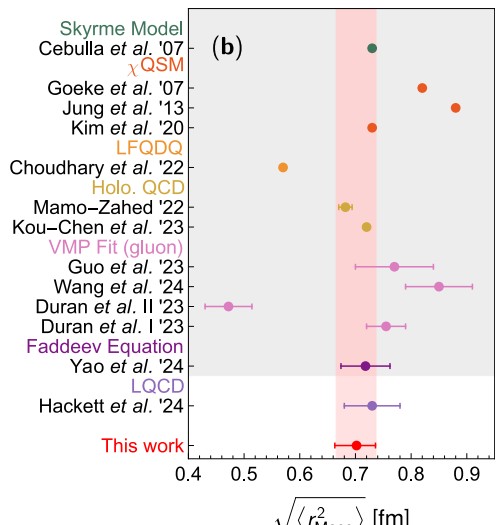

**Fig. 6 | Comparison of our results for the nucleon radius of scalar trace and mass densities with LQCD predictions[55,56]. a** The shaded region includes results from various models, including model fits to vector-meson photoproduction data[31,51,107–111], holographic QCD[29,112], $\chi$QSM[28,41], and Skyrme model[40]. **b** The shaded region includes results from various models, including Faddeev equation with the rainbow-ladder truncation[35], model fits to vector-meson photoproduction data[31,32,107], holographic QCD[29,112], light front quark-diquark model (LFQDQ)[30], chiral quark soliton model ($\chi$QSM)[28,41,44], and Skyrme model[40]. It is noted that the scale dependent results from model fits to vector-meson photoproduction data are purely gluonic.

specifically refers to the quantity derived from the energy or mass density, corresponding to $r_{Mass}$ here, while the one derived from the scalar trace density is referred to as the "scalar radius". We take the latter definition here. A comparison of our results with existing LQCD calculations and model predictions is compiled in Fig. 6. Our results agree with the LQCD results within uncertainties. Given the substantial challenges of direct measurements of GFFs, especially their gluonic components, the dispersive determinations provide invaluable insights into nucleon structure.

The nucleon radius of the scalar trace density is sizeably larger than the proton charge radius $\langle r_C^2 \rangle$, which is $\left(0.840^{+0.004}_{-0.003} \text{ fm}\right)^2$ extracted using DRs[98] and $(0.84075(64) \text{ fm})^2$ recommended by the Committee on Data of the International Science Council (CODATA)[99]. The hierarchy in radii suggests, in the sense of Wigner phase-space distribution[100,101], that gluons, which are responsible for the majority of the nucleon mass due to the trace anomaly, are distributed over a larger spatial region compared to quarks, which are responsible for the charge distribution. As a quantity characterizing gluonic dynamics in a conventional hadron, the radius of the trace density effectively represents the radius of confinement. In the MIT bag model, this radius may be considered as the bag radius[97,102], which serves as a physical boundary of confinement.

It is also instructive to show the nucleon AM[39,103] and mechanical radii[3,39,100]. The former is determined by the combination $J(t) + \frac{2}{3} t \frac{d}{dt} J(t)$ and the latter by $D(t)$, i.e., $\langle r_J^2 \rangle = 20 J'(0) = (0.70 \pm 0.2 \text{ fm})^2$, $\langle r_{Mech}^2 \rangle = \frac{6D}{\int_{-\infty}^{0} dt\, D(t)} = \left(0.72^{+0.09}_{-0.08} \text{ fm}\right)^2$. The results of various radii, together with the error budget, are given in Table 1.

The value of the mechanical radius agrees with recent LQCD results within the uncertainties[55,104,105]. The observed hierarchy of the radii corresponding to the scalar trace density, the charge distribution, and the AM distribution mirrors the hierarchy in the inverse order of the masses of the lightest mesons excited from the vacuum by the scalar gluon, vector quark-antiquark and tensor currents, respectively, which are $\sigma/f_0(500)$, $\rho(770)$, and $f_2(1270)$, respectively. The agreement in the hierarchy ordering suggests a remarkable correlation between the nucleon spatial structure and the light hadron spectrum in the

scalar, vector, and tensor channels. It is also stressed in ref. 69 that the LQCD data for the pion GFFs in ref. 75 are fully consistent with the scalar and tensor meson dominance.

## Data availability
The lattice QCD data shown in the plots were extracted from the original publications or requested from the authors of the original publication. The datasets analyzed during the current study are available from the corresponding author upon request.

## Code availability
The computer codes used to generate results are available from the corresponding author upon request.

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

## Acknowledgements

We would like to thank Jambul Gegelia for helpful discussions and the authors of ref. 55 for providing us their lattice data for Fig. 4. XHC acknowledges the School of Physics and Electronics, Hunan University, for the very kind hospitality during his stay. DLY appreciates the support of Peng Huan-Wu visiting professorship and the hospitality of Institute of Theoretical Physics at Chinese Academy of Sciences (CAS). This work was supported in part by the National Natural Science Foundation of China under Grants No. 12125507, No. 12347120, No. 12335002, No. 12375078, No. 12275076, No. 12361141819, and No. 12447101; by CAS under Grants No. YSBR-101 and No. XDB34030000; by the Postdoctoral Fellowship Program of China Postdoctoral Science Foundation under Grants No. GZC20232773 and No. 2023M74360; by the National Key R&D Program of China under Grant No. 2023YFA1606703; and by the Science Fund for Distinguished Young Scholars of Hunan Province under Grant No. 2024JJ2007. This work is also supported by the Fundamental Research Funds for the Central Universities.

## Author contributions

The authors are listed in alphabetical order. F.K.G. and D.L.Y. conceived and supervised the project. X.H.C. and Q.Z.L. performed the calculations. All authors were involved in physics discussions and in writing, editing, and reviewing the manuscript.

## Competing interests

The authors declare no competing interests.
