## [Transparent Peer Review file · Nature Communications]

Dispersive Determination of Nucleon Gravitational Form Factors

Corresponding Author: Professor Feng-Kun Guo

Version 0:

Reviewer comments:

Reviewer #1

(Remarks to the Author)

Reviewer #2

(Remarks to the Author)

This work presents an extraction of the gravitational form factors (GFFs) of the nucleon using a dispersive framework. The paper first presents the formalism necessary to obtain the pion and kaon GFFs, then extends it to the nucleon GFFs, which involve the meson results as inputs. Various free parameters appear, whose values are set using a combination of hadron masses, sum rules, results from chiral perturbation theory, etc. Besides free parameters, phase shift data from other works are taken and integrated over using dispersion relations. The resulting estimates of the GFFs are then used to compute the D-term and several different radii of the proton. The central values of these estimates compare well with estimates from model calculations and lattice QCD. Uncertainties on the free parameters are propagated to the GFFs and radii; they are smaller than those of lattice QCD estimates. The mass/scalar radius of the nucleon is found to be larger than its charge radius, which is interpreted as the spatial distribution of gluons within the nucleon being broader than that of quarks.

The work represents an ambitious and commendable attempt to advance our collective understanding about GFFs. Given the high interest in GFFs in recent years, it is highly topical. Dispersive analyses are illuminating, and their application to new quantities is nontrivial and technically challenging. However, unfortunately, we have several major concerns that we believe should be addressed before this work should be published.

1. Most importantly, the presentation lacks sufficient detail to assess whether it is technically sound and validate its results, especially the uncertainty quantification.

Unfortunately, we were simply not able to follow what was going on. We hope it will be helpful to point out some specific examples of stumbling blocks.

- When new pieces of formalism are introduced, the relation with previously established quantities is not made clear, for example, when the Omnes representation is introduced, as well as the effective zero-width poles.
- Formalism is presented interleaved with discussion of input data, but no summary is provided of the full set of input data or the final equations used to compute the main results.
- It is not clear which integrals are evaluated analytically, numerically, or a combination.
- In many places, the work leans very heavily on references. In many cases, it would be useful to be more specific about which equations (e.g. for the Omnes matrix) or what data (e.g. for the phases and phase shifts) from the outside references are used to make it easy to trace the steps.

The authors mention a forthcoming publication with details of their procedure, but without that or a Supplementary Material it is not possible from the current letter to understand how exactly several steps are performed and what are the limitations. Unfortunately, it seems likely that a significant overhaul of the presentation may be necessary to address this primary problem.

We have several secondary concerns:

2. The analysis is presented as rigorous and model-independent, but this does not seem to be appropriate. A rigorous, model-independent result should be one which can be formally guaranteed to reproduce the correct value in nature in some limit. It is not made explicit whether this is the case here (i.e., is the approach systematically improvable?). While it may be possible in principle to obtain a rigorous, model-independent result using the dispersive framework, it is clear that some approximations are being made and model dependence is entering in the analysis as implemented here. For example, tensor meson dominance is used for the determination of α , and the $\beta_{\pi/K}$ parameters are taken from NLO chPT. The present analysis will necessarily inherit the model dependence of inputs like these ones. Separately, and more importantly, some apparently ad hoc constructions enter into the formal construction. For example, in the nucleon analysis, some number of zero-width poles (how many?) are introduced to "simulate contributions from highly excited meson resonances"; while this may be reasonable, it is a model. Furthermore, it seems that the sums over intermediate states are being truncated to only the dominant contribution(s), rather than taken exhaustively (unless some aspect of the problem truncates these sums exactly which, if so, was not made clear). Can the authors provide further justification for the framing of this approach as rigorous and model-independent?

3. Although it is clear that approximations are being made, no discussion of unquantified systematic errors is provided. All of the approximations noted in the discussion of 2. above will induce some bias in the results, constituting systematic errors. However, there is no discussion in these terms. It is not made explicit where approximations or truncations are being made. Unfortunately, this may be misleading in combination with the framing as rigorous and model-independent. It needs to be made explicit where inexactnesses are entering, even when approximations are reasonable. These can be important: for example, potential missed poles are one speculated cause of the discrepancy between lattice and dispersive results for muon $g-2$. Furthermore, while full systematic error quantification is difficult and outside the scope of a first attempt, some rough estimate of expected errors should be made (with justification) wherever possible.

4. The comparison with lattice results needs more cautious framing. The paper compares the uncertainties on dispersive estimates of the GFFs with lattice results; the dispersive errorbars are smaller, which is framed as favorable. However, systematic errors are not quantified for either the dispersive or lattice results. The referenced calculations include unquantified errors due not only to the unphysical pion mass (as mentioned in this work) but also the lack of continuum/infinite volume extrapolations and others. Without such a quantification, and given the fundamental differences between the two approaches, statements about the uncertainty comparison are just not meaningful. Analogously, we could equally well write down a simple dipole form with some arbitrary choice of mass and amplitude and obtain a GFF estimate with infinitely better precision, so long as we do not quantify its systematic errors.

There are other more detailed senses in which the particular comparisons made are misaligned. For example, the authors impose sum rules for $A(0)$ and $J(0)$, which is partially why the determination of $D(0)$ is so precise compared to the lattice calculation of Ref. [55], for which the sum rules are recovered and not imposed. Imposing the sum rules on the results of that work would result in a different picture (for example, vanishing errors near $t=0$ on A and J , as seen in this work).

5. Previous works need better discussion. Extractions of GFFs from dispersive analyses have been performed in literature for the case of the pion (Ref. [70]), and for the quark D-GFF of the nucleon (Ref. [23]). Can the authors explicitly mention these previous attempts and comment on how their approach is different/improved compared to these works?

Finally, some sidenotes:

6. It should be noted explicitly that all radii presented are in the Breit frame. Moreover, in literature, the term "mass radius" is often used to refer to the one derived from the energy density, as opposed to the scalar "trace" density. What the authors call "mass radius" is in some of the literature called "scalar radius". We recommend adding a clarification to avoid confusion. Similarly, the AM radius is non-standard, and providing a definition (or even pointing to an equation in the external reference) for the density from which it is derived would be appreciated.

7. Regarding "The agreement highlights a remarkable correlation between the nucleon spatial structure and the light hadron spectrum in the scalar, vector and tensor channels.": It is not clear what exact comparison is meant; is it the ratio of values which agree, or simply the ordering? Separately, can the authors attempt an interpretation of this correlation? Could it be simply a numerical coincidence?

8. In Fig. 6, we noted at least two (Duran et al I & II 2023) radii included in the comparison which are scale dependent and derived from the purely gluonic contribution of the GFFs. If the authors want to keep comparisons against gluon-only and quark-only results in Tables 5 and 6, they should add a clarifying statement (and perhaps put those data in a differently shaded region inside the plots) to avoid confusion.

Reviewer #3

(Remarks to the Author)

Version 1:

Reviewer comments:

Reviewer #1

(Remarks to the Author)

Reviewer #2

(Remarks to the Author)

We thank the authors for their thoughtful responses to our comments and questions, and for their updates to the manuscript. In particular, we sincerely appreciate the obviously substantial efforts the authors have put into the new Supplemental Material. It is clear and informative, and its addition resolves our previous concern about the difficulties in following the formalism. Overall, the manuscript is very substantially improved. We have only a few remaining concerns, primarily pertaining to uncertainty quantification and the discussion in terms of precision.

1. We wanted to point out something curious that we noted regarding the LECs used for the pion and kaon scalar GFFs, and ask the authors to please clarify this. The LEC combination $L_{11}-L_{13}$ that is needed for the slopes of the pion and kaon GFFs is currently taken from Ref.~[2] of the Supplementary Material. Its minimum and maximum allowed values, $\$0.0003\$$ and $\$0.0007\$$, dominate the uncertainty of the calculation, as far as we can tell. In Ref.~[2], it looks like $L_{11}-L_{13}$ is obtained using Eq.~39 under the assumption that the slopes of the scalar pion and the kaon form factors at zero momentum transfer are identical. Therefore, it seems inconsistent to then use the value determined that way in order to separately compute the pion and kaon slopes. In fact, if one plugs the difference between the central values of the two that is reported in the main text, 0.02, in Eq.~39 of Ref.~[2], one finds that $L_{11}-L_{13}$ is 0.007, an order of magnitude larger than the values allowed currently, which has a significant impact on the slopes of the form factors. What is going on here, and does the uncertainty quantification need to be adjusted?

2. It seems like the truncation of the sum over intermediate states ought to be a significant source of uncertainty in this calculation. Here, a truncation to intermediate pions is considered; kaons are then added in to improve the calculation. It is not clear exactly how one would know a priori that including kaons is important, or how one would know that *only* kaons are important. Is there some structure in how one knows to add new contributions, or more importantly, some quantitative way to estimate the size of such truncation errors? This seems necessary to establish full control over uncertainties for a proper precision determination.

3. The dispersive analysis must inherit *all* of the uncertainties of its inputs. This includes whatever experimental or lattice systematics, as well as truncation uncertainties in chiral perturbation theory. It doesn't seem like χ PT truncation uncertainties are taken into account, at least.

4. There is one remaining precision comparison with lattice results, regarding $\$D(0)\$, "it agrees with recent LQCD results at $m_{\pi} = 170\$$ MeV [55], and has a considerably smaller uncertainty." We think this should be adjusted as with the other precision comparisons, as per our last comments about such comparisons not being well-posed.$

5. It seems like there are some inputs which may *only* be obtained from lattice calculations, at least with the precision required to enable a high-precision dispersive determination. Is this actually the case, or are there alternatives?

We note that most of our remaining complaints are due to the interacting issues of 1) whether full quantification of uncertainties has been achieved in this work, and 2) the framing of these results as *already* precision determinations which may be fairly compared with others. On 1), per our observations above, we don't think this has been achieved yet and might amount to substantial additional work. On 2), there are presently *no* determinations of the GFFs from any source with a comprehensive uncertainty budget, so there is not necessarily any sense in which a meaningful precision comparison could be achieved in the first place. Rather than inducing potentially unnecessary work, we want to note that essentially all of our concerns could be alleviated by some moderate reframings. On 1), whether or not this satisfies the criteria of a proper precision determination, this determination of (especially the nucleon) GFFs is technically challenging, a significant advance over the previous state of the art, and at least *lays the groundwork* for precision dispersive determinations. This by itself is more sufficient to merit publication in Nature Communications. On 2), rather than framing this method as in competition with lattice methods (given that lattice inputs are required), a better comparison may be e.g.~global fits for PDFs, GPDs, etc., which have recently been exploring including lattice inputs along with experimental data to improve precision. Reworking the paper in these terms and adding notes of any potentially large outstanding sources of uncertainty (see questions above) would address the essence of our concerns.

Reviewer #3

(Remarks to the Author)

Version 2:

Reviewer comments:

Reviewer #2

(Remarks to the Author)

We thank the authors for taking the time to address all of our comments and questions. We believe they have now all been fully addressed. We are now happy to recommend the work for publication.

Reviewer #3

(Remarks to the Author)

Reply to the referees’ reports

Xiong-Hui Cao, Feng-Kun Guo, Qu-Zhi Li and De-Liang Yao

We sincerely appreciate the positive assessments from all referees and their thorough reports and constructive comments that helped us to significantly improve our manuscript.

The main changes are outlined as follows:

1. We have improved the uncertainty quantification. Firstly, we have followed the referees’ suggestions to split various contributions to the error budget into different categories, and provided detailed numerical values (Table I in the revised version). Secondly, we have extended the mass ranges of the effective poles to include larger spectral function regions of high-lying resonances, to be more conservative. The central values and conclusions remain unchanged.
2. We have provided a Supplementary Material, which includes technical details such as the derivation of the unitarity relations, the construction of the Muskhelishvili-Omnès formalism, an explanation of the numerical methods used for handling the integrals mentioned in the main text, and a discussion on the convergence of the dispersion relations. These details allow readers to trace the steps easily. We also show plots for the input $\pi\pi$ - $K\bar{K}$ Omnès matrix and $\pi\pi/K\bar{K} \rightarrow N\bar{N}$ partial waves, as well as a prediction on the kaon trace GFF.
3. To comply with the format of Nature Communications, we have reorganized the manuscript into an Introduction section and an Results and Discussion section. We have removed the summary paragraphs and moved the last paragraph in the previous Summary section to the end of the Introduction section.

Below, we address the referees’ comments one by one and provide clarifications and explanations for the changes made in the manuscript. Our responses and modifications in the main text are highlighted in blue.

Response to Reviewer #1’s comments

In this paper, the authors employ the dispersion relations and phenomenological spectral functions in the s - and t -channels to determine the gravitational form factors (GFFs) of pion and nucleons. The imaginary parts of GFFs of pion and nucleons are dominated by $\pi\pi$, $K\bar{K}$ intermediate states, as represented in Eq. (3), (4) and (9)-(11), and higher resonances. The resulting GFFs enable the determination of the scalar (trace) radius (see comment below) and the D -term, among others. The analysis is of state-of-art in this direction and provides an alternative and well-tested approach to understand the form factors of the energy-momentum tensor. The results show good agreement with LQCD calculations at a slightly non-physical pion mass.

I will recommend the publication of this paper in Nature Communication, provided the authors address the following comments and questions satisfactorily:

1. Mass vs. scalar radii:

This paper adopted a definition of “mass radius” proposed in Ref. [50] (Ref. [51] in the revised version) through the trace or scalar form factor T_{μ}^{μ} . While this is true in the infinite-nucleon-mass limit in which the A -form factor dominates, this is not correct at the physical mass. The difference has been discussed in <https://arxiv.org/pdf/2102.07830>. More precisely, the mass should be defined through the Hamiltonian density T^{00} as the rest energy or its generalization in other Lorentz frames, having the additive property through energy. On the other hand, the matrix element of scalar/trace T_{μ}^{μ} , even though is related to mass and frame-independent, does not provide the physics of “what is mass”, and does not have the additive property. The corresponding form factor or the spatial distribution defines the scalar radius, which is related to the change of the vacuum condensate in the nucleon as well as the bag constant in the MIT model (see <https://arxiv.org/abs/2102.08191>). The authors should revise the terminology in the paper and include both the results of mass and scalar radii. Or in other words, how does dispersive analysis give different results of form factors for different operators of same quantum numbers?

Response:

We appreciate the referee for raising this key point. Indeed, there are multiple definitions of mass radius in the literature, corresponding to what should be defined as mass. According to the textbook approach [1, 2], particle mass is related to the matrix element of the trace (and the anomaly) of the energy-momentum tensor (EMT), where this trace contains both the gluon and quark contributions to mass. Alternatively, mass can be defined through the matrix element of the 00 component of the EMT as mentioned by the referee, resulting in a more refined mass decomposition. Noteworthy examples include the pioneering four-component decomposition proposed by Ji [3], among others.

Our work, as a data-driven study, does not intend to delve into the interpretation of these issues, interesting as they are, but rather focuses solely on the precise determination of the GFFs. Recognizing the difference between the two types of definitions in the literature, we have included results for both types of radii for the nucleon in the revised version. Thanks to the referee’s comment, we have changed the “mass radius” in the previous version to “radius of the scalar trace density” in the revised version (the notation was changed to $\langle r_{\Theta}^2 \rangle$). The physical results remain unchanged. The radius derived from the matrix element of T^{00} is denoted as “radius of the energy density distribution” in the revised version.

Following the referee’s suggestion, we have added the following sentence on p.6:

“The radius of the energy density distribution, derived from the matrix element of T^{00} [4], is

$$\langle r_{\text{Ener}}^2 \rangle = 6\dot{A}(0) - \frac{3D}{2m_N^2} = (0.70_{-0.04}^{+0.03} \text{ fm})^2. \quad (1)$$

”, and a comparison of our result for this radius with LQCD determination by the MIT group and results from various models in Fig. 6 (right panel).

We also added a footnote on p.6 to clarify the difference between these definitions:

“There are different definitions of the ‘mass radius’ in the literature. In Ref. [5], it is given by the radius derived from the scalar trace density, corresponding to r_{Θ} here. However, the term ‘mass radius’ in Ref. [6] specifically refers to the quantity derived from the energy density, corresponding to r_{Ener} here, while the one derived from the scalar trace density is referred to as the ‘scalar radius’.”

As for how dispersive analysis yields different results of form factors for different operators of the same quantum numbers, it can be understood from the following decomposition of the matrix element of the EMT:

$$\langle N(p')\bar{N}(p) | \hat{T}^{\mu\nu}(0) | 0 \rangle = \bar{u}(p') (T_S^{\mu\nu} + T_T^{\mu\nu}) v(p), \quad (2)$$

where the scalar and tensor parts are

$$T_S^{\mu\nu} = \frac{1}{3} \left(g^{\mu\nu} - \frac{P^\mu P^\nu}{P^2} \right) \Theta(t), \quad (3)$$

$$\begin{aligned} T_T^{\mu\nu} &= T^{\mu\nu} - \frac{1}{3} \left(g^{\mu\nu} - \frac{P^\mu P^\nu}{P^2} \right) \Theta(t) \\ &= \frac{1}{4m_N} \left[\Delta^\mu \Delta^\nu + \frac{\Delta^2}{3t} (P^\mu P^\nu - t g^{\mu\nu}) \right] A(t) + \left[i \Delta^{\{\mu} \sigma^{\nu\} \rho} P_\rho + \frac{2i \sigma^{\rho\kappa} \Delta_\rho P_\kappa}{3t} (P^\mu P^\nu - t g^{\mu\nu}) \right] J(t), \end{aligned} \quad (4)$$

While the radius of the trace density is related to the scalar part, the radius of the energy density is related to both. In the Supplementary Material, we have included the above decomposition and detailed the derivation of the discontinuity for each GFF.

2. The energy-momentum tensor cannot be uniquely identified through symmetry alone: one can easily add a term which is conserved and gives zero total charge (see Itzykson and Zuber, pg. 24). While in QCD, the symmetric and gauge invariant version of EMT is believed to couple to gravity, it is more complicated in the theory of scalars and pseudo-scalars. How does dispersive analysis (which does not refer to any specific Lagrangian or EMT operator) know which version of the EMT operators it corresponds to?

Response:

We agree with the referee that the EMT, as well as any other conserved currents, cannot be uniquely identified through symmetry alone. Dispersive analysis circumvents the ambiguity in the EMT by focusing on physical observables and their analytic properties, which are insensitive to the non-unique terms in the EMT. The uniqueness of dispersive results is established through observables or matching to chiral perturbation theory (ChPT). This approach aligns with the spirit that while a low-energy effective field theory (EFT) may not have a unique fundamental theory, the corresponding theory is determined by requiring the low-energy constants in the EFT to match values derived from the specific fundamental theory. As the referee mentioned, in QCD the symmetric and gauge invariant EMT couples to gravity. Similarly, in ChPT, the EMT represents the response of the effective action to a variation of the metric tensor [7], and since ChPT serves as a low-energy EFT of QCD, it should also be understood as coupling to gravity.

As the referee emphasized, the ambiguity in the EMT arises from adding “improved” terms that are conserved and yield zero total charge. However, these terms do not affect physical matrix elements between asymptotic states (e.g., form factors, scattering amplitudes) because they either vanish when integrated over spacetime or decouple due to their conserved nature. Dispersion relations, which are built from measurable quantities like cross sections or partial-wave amplitudes, inherently depend only on these physical matrix elements. Thus, the ambiguity in the EMT is irrelevant at the level of observables.

Another more field-theoretic perspective relies on the Källén-Lehmann spectral representation, which expresses the discontinuity of the EMT in terms of some spectral densities. The spectral density is a measurable quantity that encodes information about the physical states. Different forms of the EMT will result in the **same** spectral density if they are related by allowed “improved” terms. In other words, different EMTs related by “improved” terms produce the same spectral density because these terms contribute only to unobservable short-distance effects. Dispersive analysis extracts information from poles and cuts of the spectral densities, which are determined by the physical spectrum, not the specific local EMT operator.

In total, the dispersive analysis does not require a specific EMT operator because it works at the level of physical observables, symmetry constraints, and spectral densities. The results are free of the ambiguities in the EMT definition. Nevertheless, as the inputs are from QCD observables, the EMT operator should be understood as correspond to the one coupled to gravity.

3. It appears that the pion GFFs have already been calculated before in the literature using the dispersion approach. What is the difference between the previous results and the present one? What are the improvements, such as the error control or precision?

Response:

We thank the referee for pointing out this issue, which was not adequately addressed in our previous version. The study of pion GFFs using dispersion relations was pioneered by Donoghue, Gasser, and Leutwyler (DGL) [8]. They implemented the coupled-channel Muskhelishvili-Omnès (MO) formalism to construct the dispersion representation of the pion trace GFF. However, due to the limitations at that time, they relied on phase shifts from the CERN-Munich group in 1974 and model analyses by Au, Morgan, and Pennington in 1987, both of which suffered from low precision and presented significant challenges for error control. Additionally, their matching of the dispersion representation to ChPT at low energies relied only on LO ChPT results and some simple sum rule estimates. A recent study [9] partially improved upon the DGL work by incorporating modern phase shifts derived from solutions of the Roy equations, in addition to utilizing DGL results. Along with providing the pion trace GFF, they also employed a simple meson dominance model to fit recent lattice data by the MIT group at $m_\pi = 170$ MeV and presented results for the $A^\pi(t)$ GFF.

Our calculations of the pion GFFs also based on the MO formalism, with main differences and improvements in two aspects. Firstly, regarding the scattering phase shifts, we have adopted the Roy equation analysis results by the Bern group, which are highly precise and model-independent. Secondly, we match the dispersive representation to the NLO results of SU(3) ChPT. This improves the reliability of the dispersion representation at low energies and allows for a systematic estimation of theoretical errors via the uncertainties of the NLO ChPT low-energy constants. Moreover, we use dispersion representations for both the trace GFF Θ^π and the A^π GFF, without relying on the tensor meson dominance model directly. This approach ensures that we can give theoretical predictions on the pion and kaon GFFs without fitting to lattice data.

Consequently, our results for the pion GFFs represent the most advanced analysis within a model-independent dispersion relation framework. We also give the first dispersive prediction on the kaon trace GFF in the Supplementary Material (see Fig. 4 therein). As the low-energy constants in ChPT become more precisely determined in the future, the precision of our results can be systematically improved.

We have modified the text in the left panel of p.3, and added the following sentences in the right panel of p.3 to make these points clear:

“We note that the study of pion GFFs using the dispersion approach was pioneered in Ref. [8] with low-precision data, and further developed for Θ^π recently in Ref. [9] by incorporating S-wave $\pi\pi$ - $K\bar{K}$ scattering from dispersive analysis in Ref. [10] and fitting lattice data [11]. We advance the dispersive analysis in both Θ^π and A^π GFFs by utilizing precise phase shifts [12, 13] and NLO ChPT results [7], achieving theoretical predictions without the need for lattice data fitting.”

4. In the nucleon GFFs calculations, highly excited meson resonances are modeled using additional effective zero-width poles. This model has been widely accepted in the calculation of electromagnetic form factors. What errors do the model of spectral functions present in the result? (see the related comment below)

Response:

In low-energy hadron physics, employing dispersive analysis often encounters the issue about how to estimate the high-energy tail of the dispersion integral. In our context, this concerns the potential contributions of highly excited meson resonances above the matching point, below which phase shifts are precisely known, ~ 1.4 GeV (for quantum numbers 0^{++} and 2^{++}) to the nucleon GFFs. From a field-theoretic perspective, unitarity relation involves summing over all possible asymptotic intermediate states. In practice, however, for GFFs and other form factors, it is impossible to perform this complete sum without any approximation: higher multiparticle intermediate states

lead to more complicated singularities and there are no measurements for multiparticle scattering amplitudes. Therefore, the goal of the dispersion method is to determine the contributions of the lightest intermediate states, such as $\pi\pi$ and $K\bar{K}$, as precisely as possible.

Here, contributions from hadronic states in the energy region above ~ 1.4 GeV must be estimated using additional approximations. These contributions are expected to be suppressed significantly compared to those from the lightest intermediate states for several reasons. Firstly, contributions from multiparticle intermediate states beyond two mesons are suppressed by phase space and higher thresholds. These contributions should be negligible for the dynamics in spacelike region, unless a specific resonant enhancement mechanism exists. Secondly, the dispersion integrals converge and the high-energy contributions are intrinsically suppressed. The suppression power is controlled by the number of subtractions in the dispersion relation, naturally corresponding to the decoupling of low-energy physics from high-energy parts.

As pointed out by the referee, these contributions then need to be taken into account in the error analysis, and the zero-width pole approximation (also known as narrow-width approximation) has been widely used in the dispersive analyses of the electromagnetic form factors (see, e.g., Refs. [83–86] in the main text). In addition, this approach has also been extensively used in the recent dispersive investigations of the muon $g - 2$ (see, e.g. arxiv.org/abs/2412.00178). In summary, the use of effective poles and the error estimation (focusing on the variation of results due to changes in parameters within reasonable ranges) are well-established in dispersive analyses.

In practice, for the S-wave, the $f_0(1500)$ and $f_0(1710)$ are two significant highly excited meson resonances above 1.4 GeV, and no higher established narrow scalar resonances are known to exist (the higher $f_0(2020)$ has a width more than 400 MeV, thus its contribution is smooth and should have been covered by the extrapolation from the matching point to infinity [which is a standard procedure in dispersive analyses; see the Supplementary Material]). Since they are part of the spectral function, which enters the integrand of the dispersive integral, they only play a role in an averaged way. Thus, we use a single effective pole located in the range of $1.5 \sim 1.8$ GeV covering both resonances, with the central value 1.6 GeV, to estimate their effects (similar strategy was taken in Refs. [84,85] for accurate analyses of the nucleon electromagnetic form factors). The coupling is obtained by saturating sum rules of normalization. For the D-wave, the established highly excited meson resonances above ~ 1.4 GeV include $f_2'(1525)$, $f_2(1565)$, $f_2(1950)$, and $f_2(2010)$. Hence, we also use a single effective pole, located at $1.5 \sim 2.2$ GeV with the central value 1.8 GeV, for estimating their contributions. The explicit numerical results of uncertainties from these highly excited meson resonances are included in the revised version and shown in the error table (Table I in the main text). To be more conservative, we have increased the range of the effective pole masses. We have also carefully propagated uncertainties in the partial waves to our results. Consequently, the errors increased slightly compared to the first version.

5. The authors strongly advocate the paper presents a PRECISE determination of GFFs, as shown in the title and abstract. To qualify this, one has to have a credible estimate of errors. There is not much in the paper on a list of errors and their detailed estimations; but usually in a precision-claimed paper, an error table is needed, from which one can find what is the dominant one? how to improve in the future studies? However, I am afraid that the dispersion method cannot be made a precision tool like in chiral perturbation theory where there is a well-defined power counting. In particular, high-energy intermediate states in the spectral function could have substantial contributions. Without a systematic error analysis, it is more appropriate to replace the “precision” in the title by more descriptive “dispersive analysis”.

Response:

We sincerely thank the referee for suggesting an error table. In the revised version, we have included such a table I (Table I in the main text) and explained various sources of the errors. It can be observed that the primary sources of error stem from matching the meson GFFs to the NLO ChPT (denoted as “ChPT”) and the effective pole approximation (denoted as “eff”), while the errors resulting from the partial wave amplitudes (denoted as “pwa”) of the Roy-Steiner equations

TABLE I. Error table of D -term and radii for the corresponding nucleon GFFs. Errors in the second column are obtained by adding those in the third column in quadrature. Here “ChPT”, “pwa” and “eff” refer to the errors from the NLO ChPT inputs, the partial-wave amplitudes, and the high-energy effective poles, respectively.

D -term	$-3.38^{+0.34}_{-0.35}$	$+(0.18)_{\text{ChPT}}(0.12)_{\text{pwa}}(0.26)_{\text{eff}}$ $-(0.16)_{\text{ChPT}}(0.12)_{\text{pwa}}(0.29)_{\text{eff}}$
$\sqrt{\langle r_{\Theta}^2 \rangle}$ [fm]	$0.97^{+0.03}_{-0.03}$	$+(0.01)_{\text{ChPT}}(0.01)_{\text{pwa}}(0.03)_{\text{eff}}$ $-(0.02)_{\text{ChPT}}(0.01)_{\text{pwa}}(0.02)_{\text{eff}}$
$\sqrt{\langle r_{\text{Ener}}^2 \rangle}$ [fm]	$0.70^{+0.03}_{-0.04}$	$+(0.02)_{\text{ChPT}}(0.01)_{\text{pwa}}(0.02)_{\text{eff}}$ $-(0.02)_{\text{ChPT}}(0.01)_{\text{pwa}}(0.03)_{\text{eff}}$
$\sqrt{\langle r_{\text{Mech}}^2 \rangle}$ [fm]	$0.72^{+0.09}_{-0.08}$	$+(0.02)_{\text{ChPT}}(0.00)_{\text{pwa}}(0.09)_{\text{eff}}$ $-(0.03)_{\text{ChPT}}(0.01)_{\text{pwa}}(0.07)_{\text{eff}}$
$\sqrt{\langle r_J^2 \rangle}$ [fm]	$0.70^{+0.02}_{-0.02}$	$+(0.01)_{\text{ChPT}}(0.01)_{\text{pwa}}(0.01)_{\text{eff}}$ $-(0.01)_{\text{ChPT}}(0.00)_{\text{pwa}}(0.02)_{\text{eff}}$

are minimal.

In the revised version, we have added details about the error analysis in the second paragraph of the right panel on p.4, with the error budget presented in Table I.

We also discussed how the precision can be further improved in the future, and added the following paragraph on p.4 of the revised version:

“Nevertheless, parts of the uncertainties can be further reduced in the future. For instance, the uncertainties associated to the NLO ChPT parameters can be reduced once precise LQCD data on slopes of the pion and kaon GFFs at zero momentum transfer are available; the $\pi\pi$ scattering phase shifts up to 1.8 GeV from the very recent analysis in Ref. [14] can be used to improve the $\pi\pi$ - $K\bar{K}$ dispersive treatment beyond ~ 1.4 GeV.”

We do not share the referee’s worry “that the dispersion method cannot be made a precision tool like in chiral perturbation theory where there is a well-defined power counting”. Although the dispersion method, as an intrinsically nonperturbative method, does not have a power counting, it is based on axiomatic principles and precisely known scattering inputs (for the cases involving pions, kaons and nucleons). It covers not only the low-energy region but also regions beyond the ChPT convergence radius. Being a universal representation, the dispersive amplitudes can be matched to the ChPT ones at low energy, with the subtraction constants corresponding to combinations of low-energy constants in ChPT. With more subtractions, the dependence on high-energy inputs are also reduced. Therefore, as more low-energy data are available (either from experiments or from lattice QCD), results from the dispersive analysis can be systematically improved. In fact, the data-driven dispersive approach serves as one of the two standard methods for precise determinations of the QCD contributions to the muon $g - 2$ (the other one is lattice QCD).

Therefore, with the small errors obtained in our analysis and considering that the precision can be further improved in the future, we conclude that our analysis indeed presents a precise analysis of GFFs.

6. The physical meaning of D -term is not really clear. It certainly has nothing to do with the stability of the nucleon because quantum mechanics tells us what is stable and what is not. Its relation with chiral symmetry breaking is also a model-dependent statement. Its relation with pressure is not accurate because such a concept in a hadron is not well-defined, and moreover, pressure is a force which always involves acting and receiving parties, and it is not clear what parties are involved? These question becomes very clear in the case of hydrogen atom where D -term can also be calculated.

Response: We acknowledge the referee for raising this issue. Indeed, the physical meaning of the D -term is not really clear. Our results do not depend on any interpretation, and may serve as the basis for such studies. Therefore, we have omitted any references to mechanical stability

or pressure in the main text, and we have followed the referee's suggestion to remove the model-dependent statement ~~“it was suggested to be closely related to the spontaneous chiral symmetry breaking of QCD.”~~

We would like to thank the referee again for a careful reading and the insightful comments and suggestions. We hope that, with the above clarifications and revisions in the manuscript, the new version is suitable for publication in Nature Communications in its current form.

Response to comments in the joint report by Reviewers #2 and #3

This work presents an extraction of the gravitational form factors (GFFs) of the nucleon using a dispersive framework. The paper first presents the formalism necessary to obtain the pion and kaon GFFs, then extends it to the nucleon GFFs, which involve the meson results as inputs. Various free parameters appear, whose values are set using a combination of hadron masses, sum rules, results from chiral perturbation theory, etc. Besides free parameters, phase shift data from other works are taken and integrated over using dispersion relations. The resulting estimates of the GFFs are then used to compute the D-term and several different radii of the proton. The central values of these estimates compare well with estimates from model calculations and lattice QCD. Uncertainties on the free parameters are propagated to the GFFs and radii; they are smaller than those of lattice QCD estimates. The mass/scalar radius of the nucleon is found to be larger than its charge radius, which is interpreted as the spatial distribution of gluons within the nucleon being broader than that of quarks.

The work represents an ambitious and commendable attempt to advance our collective understanding about GFFs. Given the high interest in GFFs in recent years, it is highly topical. Dispersive analyses are illuminating, and their application to new quantities is nontrivial and technically challenging. However, unfortunately, we have several major concerns that we believe should be addressed before this work should be published.

1. Most importantly, the presentation lacks sufficient detail to assess whether it is technically sound and validate its results, especially the uncertainty quantification.

Unfortunately, we were simply not able to follow what was going on. We hope it will be helpful to point out some specific examples of stumbling blocks.

- When new pieces of formalism are introduced, the relation with previously established quantities is not made clear, for example, when the Omnès representation is introduced, as well as the effective zero-width poles.

- Formalism is presented interleaved with discussion of input data, but no summary is provided of the full set of input data or the final equations used to compute the main results.

- It is not clear which integrals are evaluated analytically, numerically, or a combination.

- In many places, the work leans very heavily on references. In many cases, it would be useful to be more specific about which equations (e.g. for the Omnès matrix) or what data (e.g. for the phases and phase shifts) from the outside references are used to make it easy to trace the steps.

The authors mention a forthcoming publication with details of their procedure, but without that or a Supplementary Material it is not possible from the current letter to understand how exactly several steps are performed and what are the limitations. Unfortunately, it seems likely that a significant overhaul of the presentation may be necessary to address this primary problem.

Response:

We really appreciate the referees for identifying the important point. Therefore, we have submitted, together with the main text, a Supplementary Material including detailed step-by-step derivations of the main formulas of this work (in particular the discontinuities of various GFFs, which are the key to constructing dispersive representations), the handling of the dispersive integrals involving the Cauchy singularity, and figures for the input $\pi\pi$ - $K\bar{K}$ coupled-channel Omnès matrix and the $\pi\pi/K\bar{K} \rightarrow N\bar{N}$ partial-wave amplitudes.

In addition, we have also added discussions in the main text, including the introduction of new formalisms and their relations to other quantities, descriptions of the input data, and a summary of the master equations used to calculate the main results etc. We have also reorganized the main text on p.4 (of the revised version) to overcome the issue “formalism is presented interleaved with discussion of input data”.

We hope that with the detailed Supplementary Material as well as the reorganization and clarifications added in the revised version of the main text, the referees (and readers) will find our work

easy to follow.

We have several secondary concerns:

2. The analysis is presented as rigorous and model-independent, but this does not seem to be appropriate.

A rigorous, model-independent result should be one which can be formally guaranteed to reproduce the correct value in nature in some limit. It is not made explicit whether this is the case here (i.e., is the approach systematically improvable?). While it may be possible in principle to obtain a rigorous, model-independent result using the dispersive framework, it is clear that some approximations are being made and model dependence is entering in the analysis as implemented here. For example, tensor meson dominance is used for the determination of α , and the $\beta_{\pi/K}$ parameters are taken from NLO ChPT. The present analysis will necessarily inherit the model dependence of inputs like these ones. Separately, and more importantly, some apparently ad hoc constructions enter into the formal construction. For example, in the nucleon analysis, some number of zero-width poles (how many?) are introduced to “simulate contributions from highly excited meson resonances”; while this may be reasonable, it is a model. Furthermore, it seems that the sums over intermediate states are being truncated to only the dominant contribution(s), rather than taken exhaustively (unless some aspect of the problem truncates these sums exactly which, if so, was not made clear). Can the authors provide further justification for the framing of this approach as rigorous and model-independent?

Response:

We agree with the referees that “a rigorous, model-independent result should be one which can be formally guaranteed to reproduce the correct value in Nature in some limit”. The accuracy of the results from such an approach should be systematically improvable.

The dispersion method is based on axiomatic principles and precisely known scattering inputs (for the cases involving pions, kaons and nucleons). Being a universal representation, the dispersive amplitudes can be matched to the ones from the chiral perturbation theory (ChPT) at low energy, which is the low-energy effective field theory of QCD. The subtraction constants correspond to combinations of low-energy constants in ChPT. With more subtractions, the dependence on high-energy inputs are also reduced. Therefore, as more low-energy data are available (either from experiments or from lattice QCD), results from the dispersive analysis can be systematically improved.

Therefore, the dispersive formalism used in this work qualifies as a model-independent approach. In fact, the data-driven dispersive approach serves as one of the two standard methods for precise determinations of the QCD contributions to the muon $g - 2$ (the other one is lattice QCD).

Now, let us discuss the approximations in our work.

Firstly, as for the tensor meson dominance used in estimating the NLO ChPT low-energy constant, indeed it is not determined from data. However, it is a good approximation as can be seen from the fact the predicted pion GFFs agree very well with the lattice results at a slightly unphysical pion mass of 170 MeV. Furthermore, to be conservative in the error estimate, we enlarge the error of the $f_2(1270)$ mass from ± 0.8 MeV averaged by the Particle Data Group to ± 20 MeV to cover various determinations from different experiments.

Secondly, we discuss the concern about zero-width poles. Since the input scattering partial waves are only precisely known up to a certain energy, all dispersive analyses encounter the issue of estimating the high-energy tail in the dispersion integral. In our context, this concerns the contributions of highly excited meson resonances above ~ 1.4 GeV (with quantum numbers 0^{++} and 2^{++}) to the nucleon GFFs.

From a field-theoretic perspective, unitarity relation involves summing over all possible asymptotic intermediate states. In practice, however, for GFFs and other form factors, it is impossible to perform this complete sum without any approximation: higher multiparticle intermediate states lead to more complicated singularities and there are no measurements for multiparticle scattering amplitudes. Therefore, the goal of the dispersion method is to determine the contributions of the

lightest intermediate states, such as $\pi\pi$ and $K\bar{K}$, in the low-energy region (up to about 1.4 GeV in our case) as precisely as possible, and contributions from high-energy region are estimated using additional approximations. These contributions are expected to be suppressed significantly compared to those from the lightest intermediate states for several reasons. Firstly, contributions from multiparticle intermediate states beyond two mesons are suppressed by phase space and higher thresholds. These contributions should be negligible for the dynamics in spacelike region, unless a specific resonant enhancement mechanism exists. Secondly, the dispersion integrals converge and the high-energy contributions are intrinsically suppressed. The suppression power is controlled by the number of subtractions in the dispersion relation, naturally corresponding to the decoupling of low-energy physics from high-energy parts.

We agree with the referees that high-energy contributions are not rigorously, precisely taken into account. Thus, such contributions need to be considered in the error analysis. The zero-width pole approximation (also known as narrow-width approximation) has been widely used in the dispersive analyses of the electromagnetic form factors (see, e.g., Refs. [83-86] in the main text). In addition, this approach has also been extensively used in the recent dispersive investigations of the muon $g - 2$ (see, e.g. arxiv.org/abs/2412.00178). In summary, the use of effective poles and the error estimation (focusing on the variation of results due to changes in parameters within reasonable ranges) are well-established in the dispersive approach.

In our context, for the S-wave, the $f_0(1500)$ and $f_0(1710)$ are two significant highly excited meson resonances above 1.4 GeV, and no higher established narrow scalar resonances are known to exist (the higher $f_0(2020)$ has a width more than 400 MeV, thus its contribution is smooth and should have been covered by the extrapolation from the matching point to infinity [which is a standard procedure in dispersive analyses; see the Supplementary Material]). Since they are part of the spectral function, which enters the integrand of the dispersive integral, they only play a role in an averaged way. Thus, we use a single effective pole located in the range of 1.5 \sim 1.8 GeV covering both resonances, with the central value 1.6 GeV, to estimate their effects (similar strategy was taken in Refs. [84,85] for accurate analyses of the nucleon electromagnetic form factors). The coupling is obtained by saturating sum rules of normalization. For the D-wave, the established highly excited meson resonances above \sim 1.4 GeV include $f_2'(1525)$, $f_2(1565)$, $f_2(1950)$, and $f_2(2010)$. Hence, we also use a single effective pole, located at 1.5 \sim 2.2 GeV with the central value 1.8 GeV, for estimating their contributions. The explicit numerical results of uncertainties from these highly excited meson resonances are included in the revised version and shown in the error table (Table I in the main text). To be more conservative, we have increased the range of the effective pole masses. We have also carefully propagated uncertainties in the partial waves to our results. Consequently, the errors increased slightly compared to the first version.

Therefore, based on axiomatic principles, with the high-energy contributions included in the error estimates, and improvable upon new experimental and lattice data, our formalism qualifies as a rigorous, model-independent approach.

3. Although it is clear that approximations are being made, no discussion of unquantified systematic errors is provided.

All of the approximations noted in the discussion of 2. above will induce some bias in the results, constituting systematic errors. However, there is no discussion in these terms. It is not made explicit where approximations or truncations are being made. Unfortunately, this may be misleading in combination with the framing as rigorous and model-independent. It needs to be made explicit where inexactnesses are entering, even when approximations are reasonable. These can be important: for example, potential missed poles are one speculated cause of the discrepancy between lattice and dispersive results for muon $g - 2$. Furthermore, while full systematic error quantification is difficult and outside the scope of a first attempt, some rough estimate of expected errors should be made (with justification) wherever possible.

Response:

We appreciate the referee highlighting this issue. In the revised version, we have included a

detailed error table I (Table I in the main text) and briefly explained the sources of error. It can be observed that the primary sources of error stem from matching the meson GFFs to the NLO ChPT (denoted as “ChPT”) and the effective pole approximation (denoted as “eff”), while the errors resulting from the partial wave amplitudes (denoted as “pwa”) of the Roy-Steiner equations are minimal.

To make the point clear, we have rewritten the uncertainty analysis in the main text as follows: *“The uncertainties of our results come from three sources: (i) uncertainties of the LECs in NLO ChPT [7], which are obtained by varying $\alpha \in [-0.03, 0.01] \text{ GeV}^{-2}$, $\beta_\pi \in [0.68, 0.72]$ and $\beta_K \in [0.32, 0.60]$, corresponding to varying L_{12}^r , $\dot{\Theta}^\pi(0)$ and $\dot{\Theta}^K(0)$ as given above in the mesonic sector; (ii) uncertainties of the $\pi\pi/K\bar{K} \rightarrow N\bar{N}$ partial wave amplitudes, which have been fully estimated in the comprehensive review of the πN Roy-Steiner equation analysis [15]; (iii) uncertainties of the high-energy tail of the spectral functions, estimated by varying the effective pole masses. In practice, for the S-wave, we use one effective pole located at $1.5 \sim 1.8 \text{ GeV}$ with the central value 1.6 GeV to cover both the $f_0(1500)$ and $f_0(1710)$ resonances; for the D-wave, we use one effective pole located at $1.5 \sim 2.2 \text{ GeV}$ with the central value 1.8 GeV to cover the $f_2(1565)$, $f_2(1950)$ and $f_2(2010)$ resonances. The above error budget is summarized in Table. I, where the three different sources of uncertainties are denoted as “ChPT”, “pwa” and “eff”, respectively.*

Nevertheless, parts of the uncertainties can be further reduced in the future. For instance, the uncertainties associated to the NLO ChPT parameters can be reduced once precise LQCD data on slopes of the pion and kaon GFFs at zero momentum transfer are available; the $\pi\pi$ scattering phase shifts up to 1.8 GeV from the very recent analysis in Ref. [14] can be used to improve the $\pi\pi$ - $K\bar{K}$ dispersive treatment beyond $\sim 1.4 \text{ GeV}$. ”

4. The comparison with lattice results needs more cautious framing.

The paper compares the uncertainties on dispersive estimates of the GFFs with lattice results; the dispersive errorbars are smaller, which is framed as favorable. However, systematic errors are not quantified for either the dispersive or lattice results. The referenced calculations include unquantified errors due not only to the unphysical pion mass (as mentioned in this work) but also the lack of continuum/infinite volume extrapolations and others. Without such a quantification, and given the fundamental differences between the two approaches, statements about the uncertainty comparison are just not meaningful. Analogously, we could equally well write down a simple dipole form with some arbitrary choice of mass and amplitude and obtain a GFF estimate with infinitely better precision, so long as we do not quantify its systematic errors.

There are other more detailed senses in which the particular comparisons made are misaligned. For example, the authors impose sum rules for $A(0)$ and $J(0)$, which is partially why the determination of $D(0)$ is so precise compared to the lattice calculation of Ref. [55], for which the sum rules are recovered and not imposed. Imposing the sum rules on the results of that work would result in a different picture (for example, vanishing errors near $t = 0$ on A and J , as seen in this work).

Response:

As we mentioned above, we have quantified the systematic errors and separated various error sources in an error table. We understand the referees’ concern on unquantified systematic errors in lattice calculations, such as the lack of continuum and infinite volume extrapolations, among others. In fact, these systematic errors have been roughly estimated in the Supplementary Material of the lattice paper, Ref. [16], suggesting that they could affect the overall normalization of the GFFs by about 20%, while the t -dependence is less affected by these effects.

A more detailed analysis, such as the effects of chiral extrapolation from unphysical pion masses to physical mass point, is indeed interesting and important; in fact we have been working on this issue. Although we do not think the difference between the results at the physical pion mass and at a 170 MeV pion mass would be sizable, we remove the wordings on the error comparisons in the revised version of the main text.

However, regarding the comparison of $D(0)$, there seems to be some misunderstanding. While we indeed provide normalization constraints using sum rules for the GFFs $A(0)$, $J(0)$, and $\Theta(0)$, it

is important to note that these constraints do not directly apply to $D(0)$. By utilizing a relation satisfied by the GFFs, $\Theta(t) = m_N \left[A(t) - \frac{t}{4m_N^2} (A(t) - 2J(t) + 3D(t)) \right]$ (Eq.(2) of the main text, which was an inline equation in the previous version), one can take the derivative respect to t and then set $t = 0$,

$$4m_N \Theta'(0) = 4m_N^2 A'(0) - A(0) + 2J(0) - 3D(0) = 4m_N^2 A'(0) - 3D(0) . \quad (5)$$

It follows that, $D(0)$ is only related to the derivatives of the GFFs A and Θ at $t = 0$,

$$D(0) = \frac{4m_N}{3} (m_N A'(0) - \Theta'(0)) . \quad (6)$$

A different sum rule

$$D(0) = \frac{4m_N}{3\pi} \int_{4m_\pi^2}^{\infty} dt' \frac{\text{Im}(m_N A(t') - \Theta(t'))}{t'^2} . \quad (7)$$

can be derived from the dispersion relations

$$(A, \Theta)(t) = \frac{1}{\pi} \int_{4m_\pi^2}^{\infty} dt' \frac{\text{Im}(A, \Theta)(t')}{t' - t} . \quad (8)$$

Compared to sum rules of normalization, this sum rule converges faster and is less sensitive to the high-energy tails due to the suppression factor $1/t'^2$ rather than $1/t'$ at large t' (cf. sum rules of normalization). It explains why $D(0)$ is more precise compared to lattice results. The above equalities are added as Eq. (17) in the main text, and the detailed derivations (though straightforward) can be found in the Supplementary Material.

In addition, lattice calculations evaluate the quark and gluon contributions to the GFFs separately, which makes it challenging to impose normalization constraints on the overall GFFs. Moreover, the feature of lattice calculations makes it challenging to control uncertainties at low $|t|$ [16]. However, one of the key advantages of dispersive analysis is its capability to produce precise results at low $|t|$.

5. Previous works need better discussion.

Extractions of GFFs from dispersive analyses have been performed in literature for the case of the pion (Ref. [70] (Ref. [69] in the revised version), and for the quark D-GFF of the nucleon (Ref. [23]). Can the authors explicitly mention these previous attempts and comment on how their approach is different/improved compared to these works?

Response:

The study of pion GFFs using dispersion relations was pioneered by Donoghue, Gasser, and Leutwyler (DGL) [8]. They implemented the coupled-channel Muskhelishvili-Omnès (MO) formalism to construct the dispersion representation of the pion trace GFF. However, due to the limitations at that time, they relied on phase shifts from the CERN-Munich group in 1974 and model analyses by Au, Morgan, and Pennington in 1987, both of which suffered from low precision and presented significant challenges for error control. Additionally, their matching of the dispersion representation to ChPT at low energies relied only on LO ChPT results and some simple sum rule estimates. A recent study [9] (Ref.[68] mentioned by the referee) partially improved upon the DGL work by incorporating modern phase shifts derived from solutions of the Roy equations, in addition to utilizing DGL results. Along with providing the pion trace GFF, they also employed a simple tensor meson dominance model, saturated with the $f_2(1270)$ meson, to fit recent lattice data by the MIT group at $m_\pi = 170$ MeV and present results for the $A^\pi(t)$ GFF.

Our calculations of the pion GFFs are also based on the MO formalism, with main differences and improvements in two aspects. Firstly, regarding the scattering phase shifts, we have adopted the

Roy equation analysis results by the Bern group, which are highly precise and model-independent. Secondly, we match the dispersive representation to the NLO results of SU(3) ChPT. This improves the reliability of the dispersion representation at low energies and allows for a systematic estimation of theoretical errors via the uncertainties of the NLO ChPT low-energy constants. Moreover, we use dispersion representations for both the trace GFF Θ^π and the A^π GFF, without relying on the tensor meson dominance model directly. This approach ensures that we can give theoretical predictions on the pion and kaon GFFs without fitting to lattice data.

As for the quark D-GFF of the nucleon, Ref. [17] provides a dispersion representation for the quark contribution to the quark D -term form factor, by using hard exclusive reactions such as deeply virtual Compton scattering. The unitarity relation is saturated with only two-pion intermediate states. The imaginary part inputs for the dispersion relations are obtained from two-pion generalized distribution amplitudes (GDAs), which are calculated through dispersion relations using the Omnès representation, and the partial waves for the $\pi\pi \rightarrow N\bar{N}$ amplitudes obtained from the analytic continuation of πN scattering based on the old Karlsruhe-Helsinki (KH) dispersive analysis [18].

Their work is inspiring, and the dispersive framework is, in principle, model-independent. However, certain aspects are less satisfactory. Firstly, their results rely on estimates of the 2π GDA. This is very challenging because 2π GDA is related to the parton distribution functions of the pion. Secondly, they only considered the 2π intermediate states, neglecting $K\bar{K}$ intermediate states despite the significant interactions between S -wave $\pi\pi$ and $K\bar{K}$ around 1 GeV. Furthermore, the $\pi\pi \rightarrow \bar{N}N$ KH partial wave amplitudes used there are limited to $t < 0.78$ GeV². Last but not least, an error estimate was missing.

Our work considers not only the D -term GFF, but also other GFFs of the nucleon. All of them sum up both quark and gluon contributions and are scale independent. We include both the S -wave $\pi\pi$ and $K\bar{K}$ coupled channels, as well as effective poles to approximate the high-energy contributions above ~ 1.4 GeV. We also utilize the results from the latest πN Roy-Steiner equations, which provide model-independent $\pi\pi \rightarrow N\bar{N}$ partial wave amplitudes. Additionally, we have provided an analysis of the systematic uncertainties (improved in the revised version by providing an error table).

Consequently, our results for the pion and nucleon GFFs represent the most advanced results within a model-independent dispersion relation framework. We also give the first dispersive prediction on the kaon trace GFF in the Supplementary Material (see Fig. 4 therein). As the low-energy constants in ChPT become more precisely determined in the future, the precision of our results can be systematically improved.

We have added the following paragraph on p.3:

“We note that the study of pion GFFs using the dispersion approach was pioneered in Ref. [8] with low-precision data, and further developed for Θ^π recently in Ref. [9] by incorporating S -wave $\pi\pi$ - $K\bar{K}$ scattering from dispersive analysis in Ref. [10] and fitting lattice data [11]. We advance the dispersive analysis in both Θ^π and A^π GFFs by utilizing precise phase shifts [12, 13] and NLO ChPT results [7], achieving theoretical predictions without the need for lattice data fitting.”

and the following from the end of p.4 to p.5:

“It is noted that Ref. [17] offers a dispersive analysis for the quark D -term GFF of the nucleon in deeply virtual Compton scattering. This pioneering work is limited in several aspects: model dependent estimate of the 2π generalized distribution amplitudes, neglect of the $K\bar{K}$ intermediate state and the absence of an error analysis. These limitations have been overcome in our work, which offers the first dispersive determination of all nucleon GFFs, by incorporating S -wave $\pi\pi$ - $K\bar{K}$ coupled channels, using the partial waves from the modern πN Roy-Steiner equation analysis instead of old Karlsruhe-Helsinki results [18], and offering a reasonable estimate of uncertainties.”

Finally, some sidenotes:

6. It should be noted explicitly that all radii presented are in the Breit frame. Moreover, in

literature, the term “mass radius” is often used to refer to the one derived from the energy density, as opposed to the scalar “trace” density. What the authors call “mass radius” is in some of the literature called “scalar radius”. We recommend adding a clarification to avoid confusion. Similarly, the AM radius is non-standard, and providing a definition (or even pointing to an equation in the external reference) for the density from which it is derived would be appreciated.

Response:

We agree with the referees that all radii presented are in the Breit frame. This has been specified in the revised version (2nd paragraph on p.6).

Recognizing the distinction between the two types of definitions of the “mass radius” in the literature, we have changed the terminology to the unambiguous “radius of the scalar trace density” and, correspondingly, the notation from $\langle r_{\text{Mass}}^2 \rangle$ to $\langle r_{\Theta}^2 \rangle$. Moreover, we have also given results on the radius of the energy density distribution, denoted as $\langle r_{\text{Ener}}^2 \rangle$.

To make the AM radius more clear, we have cited the references for the AM and mechanical radii separately and added the corresponding AM density $J(t) + \frac{2}{3}t\frac{d}{dt}J(t)$ before the numerical result (bottom of the left panel on p.6).

7. Regarding “The agreement highlights a remarkable correlation between the nucleon spatial structure and the light hadron spectrum in the scalar, vector and tensor channels.”: It is not clear what exact comparison is meant; is it the ratio of values which agree, or simply the ordering? Separately, can the authors attempt an interpretation of this correlation? Could it be simply a numerical coincidence?

Response: The comparison refers to the ordering. To stress this point, we have added “in the hierarchy ordering” after “The agreement” in the revised version. We believe that the agreement between the ordering of the radii with the ordering of the inverse scalar, vector and tensor meson masses is not a numerical coincidence although we are not able to prove the statement. To reflect this point, we changed the word “highlights” to “suggests” in the revised version. We also notice that main conclusion of Ref. [9] (Ref. [69] in the revised version) is that the MIT lattice data for the pion GFFs are fully consistent with the scalar and tensor meson dominance. A sentence on this point was added before the Summary section.

8. In Fig. 6, we noted at least two (Duran et al I & II 2023) radii included in the comparison which are scale dependent and derived from the purely gluonic contribution of the GFFs. If the authors want to keep comparisons against gluon-only and quark-only results in Figs. 5 and 6, they should add a clarifying statement (and perhaps put those data in a differently shaded region inside the plots) to avoid confusion.

Response:

We have added the following sentence in the caption of Fig. 6 in the main text to avoid confusion: *“It is noted that the scale dependent results from model fits to vector-meson photoproduction data are purely gluonic.”*

We also added in the caption of Fig. 4 in the main text that the LQCD result from Ref. [56] is purely gluonic.

We would like to thank the referees again for their careful reading and the insightful comments and suggestions. We hope that, with the above clarifications, the detailed Supplementary Material and revisions in the manuscript, the new version is suitable for publication in Nature Communications in its current form.

-
- [1] M. A. Shifman, A. I. Vainshtein, and V. I. Zakharov, *Phys. Lett. B* **78**, 443 (1978).
 - [2] J. F. Donoghue, E. Golowich, and B. R. Holstein, *Dynamics of the Standard Model: Second edition* (Cambridge University Press, 2022).
 - [3] X.-D. Ji, *Phys. Rev. Lett.* **74**, 1071 (1995), arXiv:hep-ph/9410274.
 - [4] M. V. Polyakov and P. Schweitzer, *Int. J. Mod. Phys. A* **33**, 1830025 (2018), arXiv:1805.06596 [hep-ph].
 - [5] D. E. Kharzeev, *Phys. Rev. D* **104**, 054015 (2021), arXiv:2102.00110 [hep-ph].
 - [6] X. Ji, *Front. Phys. (Beijing)* **16**, 64601 (2021), arXiv:2102.07830 [hep-ph].
 - [7] J. F. Donoghue and H. Leutwyler, *Z. Phys. C* **52**, 343 (1991).
 - [8] J. F. Donoghue, J. Gasser, and H. Leutwyler, *Nucl. Phys. B* **343**, 341 (1990).
 - [9] W. Broniowski and E. Ruiz Arriola, *Phys. Lett. B* **859**, 139138 (2024), arXiv:2405.07815 [hep-ph].
 - [10] A. Celis, V. Cirigliano, and E. Passemar, *Phys. Rev. D* **89**, 013008 (2014), arXiv:1309.3564 [hep-ph].
 - [11] D. C. Hackett, P. R. Oare, D. A. Pefkou, and P. E. Shanahan, *Phys. Rev. D* **108**, 114504 (2023), arXiv:2307.11707 [hep-lat].
 - [12] B. Ananthanarayan, G. Colangelo, J. Gasser, and H. Leutwyler, *Phys. Rept.* **353**, 207 (2001), arXiv:hep-ph/0005297.
 - [13] P. Bydžovský, R. Kamiński, and V. Nazari, *Phys. Rev. D* **94**, 116013 (2016), arXiv:1611.10070 [hep-ph].
 - [14] J. R. Peláez, P. Rabán, and J. Ruiz de Elvira, *Phys. Rev. D* **111**, 074003 (2025).
 - [15] M. Hoferichter, J. Ruiz de Elvira, B. Kubis, and U.-G. Meißner, *Phys. Rept.* **625**, 1 (2016), arXiv:1510.06039 [hep-ph].
 - [16] D. C. Hackett, D. A. Pefkou, and P. E. Shanahan, *Phys. Rev. Lett.* **132**, 251904 (2024), arXiv:2310.08484 [hep-lat].
 - [17] B. Pasquini, M. V. Polyakov, and M. Vanderhaeghen, *Phys. Lett. B* **739**, 133 (2014), arXiv:1407.5960 [hep-ph].
 - [18] G. Höhler, *Pion-Nukleon-Streuung: Methoden und Ergebnisse phänomenologischer Analysen. Teil 2*, in Landolt-Börnstein 9b2, H. Schopper eds., Springer Verlag, Berlin Germany (Springer, 1983).

Reply to the referees’ second reports

Xiong-Hui Cao, Feng-Kun Guo, Qu-Zhi Li and De-Liang Yao

We sincerely appreciate the positive assessments from all referees and their thorough reports and constructive comments. Below, we address the referees’ comments one by one and provide clarifications and explanations for the changes made in the manuscript. Our responses and modifications in the main text are highlighted in blue.

Response to Reviewer #1’s comments

The authors have done extensive revisions on the first version submitted, included a table of uncertainties and a Supplementary Materials which is very helpful in understanding some of technical details. Moreover, there are substantial changes in the text which reflected comments from previous round of reports. I am generally happy now with the paper, and recommend its publication with some optional revisions which can take into account my additional comments and feedback below. I don’t need to see the revised version again, which shall be directly go to publication.

Here are some further comments:

1. Mass radius cannot be defined from the trace of EMT. This is wrong, because trace of EMT is not a mass operator as was emphasized in my first report. In all models of the nucleon, the mass is calculated from Hamiltonian as energy, same in lattice QCD calculations where two-point correlator at large Euclidean time is related to the energy of the system. In covariant perturbation theory, calculations of mass are identified by poles of Green’s functions. This pole is invariant but when the momentum of the particle is zero, it is the rest energy. Masses must be ultimately related to scalars of a theory, like trace of EMT, or Higgs vev, but they are NOT mass directly. Therefore, there is only one mass radius which is the energy radius (Einstein said, ”mass is energy”). If you prefer calling it energy radius, this is fine, but mass radius definitely sounds better because gravity is generally considered as from mass in non-relativistic limit.

Response:

We appreciate the referee for highlighting this conceptual point again. Now we completely agree with the insightful discussion about the definition of mass radius. In this further revised version, we have changed “energy radius” to “mass radius”, which was in fact also suggested in the first report by the other referees.

2. My comment about the form factors of different operators with the same quantum numbers was not understood correctly. So let me ask in a different way. I can construct another second-rank tensor in QCD or meson theory. For example, I can multiply the EMT operator with scalar gluon fields FF to get another operator with the same quantum number. Surely, its form factor as a function of t will be very different. But your method seems to suggest they will have the same form factors (of course, the new operator is not conserved, but one can correct this by adding some additional term). If so, how does one know your result on t dependence is for the form factors of the very EMT? This question will not impact the result of your paper, but is interesting to consider.

Response:

This is an interesting yet deep question. We take this opportunity to offer some crucial aspects missed in the previous reply.

Let us consider the discontinuity of a form factor for an operator (taking a scalar operator and scalar particles for illustration). Considering only two-body intermediate states, we obtain the following unitarity relation:

$$\text{disc } F_a(s) = 2i \text{Im } F_a(s) = 2i \sum_b F_b(s) \rho_b(s) T_{ba}^*(s) \theta(\sqrt{s} - m_{b1} - m_{b2}), \quad (1)$$

where we use a and b to label the two-particle channels, $\rho_b(s)$ is the two-body phase space factor, and $T_{ba}(s)$ is the S -wave scattering amplitude between channel- b and channel- a . One sees that the part of $2i\rho_b(s)T_{ba}^*(s)\theta(\sqrt{s} - m_{b1} - m_{b2})$ in the summation is universal for all such form factors. However, since the above equation is homogeneous, one can multiply an arbitrary holomorphic (and thus does not have any discontinuity) function to both sides, which is not constrained by unitarity. Since the high-energy asymptotic behavior of a form factor is constrained to drop with a negative power of Q^2 (up to a logarithmic factor) by perturbative QCD, the holomorphic function should be a polynomial up to a given order. Thus, the answer to the question is that the form factors of different operators with the same quantum numbers can differ by overall factors that are polynomials, which may be further constrained by low-energy symmetries. In particular, for coupled-channel systems, these factors determine the relative contributions of different channels and can lead to very different t dependence as pointed out by the referee.

We illustrate the above point with one pertinent example. The pion trace GFF $\Theta^\pi(t)$ and the pion σ -term form factor,

$$\langle \pi^a(p') \pi^b(p) | m_u \bar{u}u + m_d \bar{d}d | 0 \rangle = \delta^{ab} \Gamma^\pi(t), \quad (2)$$

share the same quantum numbers (0^{++}), both being scalar matrix elements. The distinction between these two different matrix elements is based on their different low-energy properties, as they satisfy different low-energy theorems. For the σ -term form factor, its dispersive representation takes the form [1]

$$\begin{aligned} \Gamma^\pi(t) &= \Gamma^\pi(0) (\mathbf{\Omega}_0^0)_{11}(t) + \frac{2}{\sqrt{3}} \Gamma^K(0) (\mathbf{\Omega}_0^0)_{12}(t), \\ &\stackrel{\text{LO}}{=} m_\pi^2 (\mathbf{\Omega}_0^0)_{11}(t) + \frac{1}{\sqrt{3}} m_\pi^2 (\mathbf{\Omega}_0^0)_{12}(t) \propto m_\pi^2. \end{aligned} \quad (3)$$

However, for the pion trace GFF, it does not vanish in the chiral limit when the pseudo-Nambu-Goldstone bosons become massless, indicating that the trace GFF cannot be directly proportional to the meson masses. Its explicit expression is [1]

$$\Theta^\pi(t) = (2m_\pi^2 + \beta_\pi t) (\mathbf{\Omega}_0^0)_{11}(t) + \frac{2}{\sqrt{3}} (2m_K^2 + \beta_K t) (\mathbf{\Omega}_0^0)_{12}(t), \quad (4)$$

where the parameters β_π, β_K are related to the slopes at $t = 0$,

$$\begin{aligned} \beta_\pi &= \dot{\Theta}^\pi(0) - 2m_\pi^2 \left(\dot{\mathbf{\Omega}}_0^0 \right)_{11}(0) - \frac{4m_K^2}{\sqrt{3}} \left(\dot{\mathbf{\Omega}}_0^0 \right)_{12}(0), \\ \beta_K &= \dot{\Theta}^K(0) - \sqrt{3}m_\pi^2 \left(\dot{\mathbf{\Omega}}_0^0 \right)_{21}(0) - 2m_K^2 \left(\dot{\mathbf{\Omega}}_0^0 \right)_{22}(0). \end{aligned} \quad (5)$$

The slopes $\dot{\Theta}^{\pi,K}(0)$ can be calculated in ChPT in terms of low-energy constants, as discussed in the Supplementary Material. This shows how the form factors of operators with the same quantum numbers differ and how the polynomial difference is constrained by low-energy chiral symmetry.

3. I think the reply about non-uniqueness of EMT is not entirely correct. Your result might be the one that couples to gravity, but there are other possible forms of EMT which will have different form factors. There is no proof that the superpotential term vanishes in a physical eigenstate. This is related to the comment above. This question will not impact the result of your paper, but is interesting to consider.

Response:

We agree with you that there are other possible forms of the EMT that will have different form factors, which will differ by polynomials as discussed in the reply to the above point. Here, we focus exclusively on the GFF of mesons and nucleons within the framework of QCD, without involving other complex issues. Again let us take the pion trace GFF as an example. The polynomial ambiguity in our context is fixed by matching the low-energy behavior of the GFF to the results in ChPT; see Ref. [2] for a recent discussion about the definition of the GFF in ChPT.

4. I am happy with the reply about the previous literature on pion EMT form factors.

Response:

Thanks.

5. If looking at the trace anomaly operator, which is FF in chiral limit, I would guess scalar and tensor glueballs will make very significant contributions in the spectral function. I would not know how to quantify errors on these unknowns. I will allow to use the precision in title, but I can point out still other places one does not have control. One such place is Eq. (35) in the Supplement: without additional argument, simply take a linear form is a type of model for polynomials (maybe asymptotic large t analysis will help here).

Response:

We agree with the referee that studying the trace GFF in the chiral limit would be interesting. However, we believe that in the realistic case, there should not be additional glueball contributions beyond what we have already considered. The key point is that in our dispersive approach based on unitarity, only QCD asymptotic states can appear as intermediate states. All resonance information is encoded in the scattering amplitudes ($\pi\pi \rightarrow \pi\pi$, $\pi\pi \rightarrow K\bar{K}$, $\pi\pi/K\bar{K} \rightarrow N\bar{N}$, etc.). In the large N_c limit, glueballs would need to be included separately as zero-width poles due to their stability; however, this is not the case in realistic QCD. In realistic QCD, pure glueball states do not exist, and any states with significant glueball components—if they exist at all—are already incorporated into the aforementioned scattering amplitudes. Nevertheless, glueball-dominant states may exist above ~ 1.5 GeV, and such high-energy contributions are partially captured by our effective poles. Although these high-energy contributions should be suppressed due to the convergence of the twice-subtracted dispersive integral, they are indeed modeled. Thus, to be more conservative, we have decided to change “Precision” to the more narrative “Dispersive” in the title.

Regarding Eq. (35) in the Supplement, the polynomial stems from the polynomial ambiguity of the dispersive formalism, as mentioned above. We employ a linear function to match the low-energy behavior in ChPT at NLO. Thus, this aspect is not a model assumption, but rather a consequence of the chiral expansion at low energies.

6. The physics of a large scalar radius is clearly seen in MIT bag model where bag radius (or confinement radius) is related to this, and the bag constant is the QCD trace anomaly. The relevant physics has been commented in Ref. [104], and also in <https://arxiv.org/abs/2105.03974>. But I don't think Ref. [107] and Ref. [108] are relevant.

Response:

We are grateful to the referee for reminding us of this intriguing point and for the references. Thinking along the line, indeed the scalar radius may be regarded as a confinement radius for a conventional hadron. We have added in the main text the following sentences: *“As a quantity characterizing gluonic dynamics in a conventional hadron, the radius of the trace density effectively*

represents the radius of confinement. In the MIT bag model, this radius may be considered as the bag radius [104,109], which serves as a physical boundary of confinement.” where [109] is the preprint mentioned in the comment. We have also added a sentence in the abstract “*indicating that the radius of the trace density may be regarded as a confinement radius.*”

We would like to thank the referee again for careful reading and insightful comments and suggestions. We hope that, with the above clarifications and revisions in the manuscript, the new version is suitable for publication in Nature Communications in its current form.

-
- [1] J. F. Donoghue, J. Gasser, and H. Leutwyler, Nucl. Phys. B **343**, 341 (1990).
[2] H. Alharazin, D. Djukanovic, J. Gegelia, and M. V. Polyakov, Phys. Rev. D **102**, 076023 (2020), arXiv:2006.05890 [hep-ph].

Response to comments in the joint report by Reviewers #2 and #3

We thank the authors for their thoughtful responses to our comments and questions, and for their updates to the manuscript. In particular, we sincerely appreciate the obviously substantial efforts the authors have put into the new Supplemental Material. It is clear and informative, and its addition resolves our previous concern about the difficulties in following the formalism. Overall, the manuscript is very substantially improved. We have only a few remaining concerns, primarily pertaining to uncertainty quantification and the discussion in terms of precision.

1. We wanted to point out something curious that we noted regarding the LECs used for the pion and kaon scalar GFFs, and ask the authors to please clarify this. The LEC combination $L_{11} - L_{13}$ that is needed for the slopes of the pion and kaon GFFs is currently taken from Ref. [2] of the Supplementary Material. Its minimum and maximum allowed values, 0.0003 and 0.0007, dominate the uncertainty of the calculation, as far as we can tell. In Ref. [2], it looks like $L_{11} - L_{13}$ is obtained using Eq. (39) under the assumption that the slopes of the scalar pion and the kaon form factors at zero momentum transfer are identical. Therefore, it seems inconsistent to then use the value determined that way in order to separately compute the pion and kaon slopes. In fact, if one plugs the difference between the central values of the two that is reported in the main text, 0.02, in Eq. (39) of Ref. [2], one finds that $L_{11} - L_{13}$ is 0.007, an order of magnitude larger than the values allowed currently, which has a significant impact on the slopes of the form factors. What is going on here, and does the uncertainty quantification need to be adjusted?

Response:

There appears to be some misunderstanding. While it is true that Eq. (39) in Ref. [2] of the Supplementary Material can be used to determine the combination $L_{11}^r - L_{13}^r$, it is not the only method for estimating these LECs. In addition to Eq. (39),

$$\begin{aligned} & \dot{\Theta}^\pi(0) - \dot{\Theta}^K(0) \\ &= \frac{4(m_K^2 - m_\pi^2)}{F_\pi^2} \left\{ 6L_{11}^r + L_{12}^r - 6L_{13}^r - \frac{1}{192\pi^2} \left(\ln \frac{m_\eta^2}{\mu^2} + 1 \right) \right\} - \frac{m_\pi^2}{32\pi^2 F_\pi^2} \ln \frac{m_\eta^2}{m_\pi^2}, \end{aligned} \quad (6)$$

Ref. [2] also employed the meson dominance method to estimate these LECs, and by combining the two approaches, a reasonable error estimate was provided. The difference between the slopes of the scalar pion and kaon trace GFFs at zero momentum transfer $t = 0$ is consistent with the error estimate. There are multiple ways to cross-check this assertion. For instance, if we use the central value of $\dot{\Theta}^\pi(0) - \dot{\Theta}^K(0) = 0.04$ as applied in our main text ($\dot{\Theta}^\pi(0) = 0.98(2)$, $\dot{\Theta}^K(0) = 0.94(14)$) and substitute it into the above equation—Eq. (39) in Ref. [2], we obtain

```

In[51]:= NSolve[
数值求解
  
$$\frac{4(0.496^2 - 0.14^2)}{0.092^2} \left( 6x + (-2.7 \times 10^{-3}) - \frac{1}{192 \text{ Pi}^2} \left( \text{Log} \left[ \frac{0.548^2}{1.0^2} \right] + 1 \right) \right) - \frac{0.14^2}{32 \text{ Pi}^2 * 0.092^2} \text{Log} \left[ \frac{0.548^2}{0.14^2} \right] == 0.04, x]
对数
Out[51]= {{x -> 0.000525623}}$$

```

where $F_\pi = 0.092$ MeV, $L_{12}^r = -\frac{F_\pi^2}{2m_{f_2}^2} = -2.7 \times 10^{-3}$ and $\mu = 1$ GeV are taken from Ref. [2]. The result is just in the middle of the uncertainty range from 0.0003 to 0.0007. Alternatively, using values of 0.02 or 0 for $\dot{\Theta}^\pi(0) - \dot{\Theta}^K(0)$ results in

$$\text{In[53]:= NSolve}\left[\frac{4(0.496^2 - 0.14^2)}{0.092^2} \left(6x + (-2.7 * 10^{-3}) - \frac{1}{192 \text{Pi}^2} \left(\text{Log}\left[\frac{0.548^2}{1.0^2}\right] + 1\right)\right) - \frac{0.14^2}{32 \text{Pi}^2 * 0.092^2} \text{Log}\left[\frac{0.548^2}{0.14^2}\right] = 0.02, x\right]$$

Out[53]= {{x -> 0.000494471}}

$$\text{In[52]:= NSolve}\left[\frac{4(0.496^2 - 0.14^2)}{0.092^2} \left(6x + (-2.7 * 10^{-3}) - \frac{1}{192 \text{Pi}^2} \left(\text{Log}\left[\frac{0.548^2}{1.0^2}\right] + 1\right)\right) - \frac{0.14^2}{32 \text{Pi}^2 * 0.092^2} \text{Log}\left[\frac{0.548^2}{0.14^2}\right] = 0.0, x\right]$$

Out[52]= {{x -> 0.000463319}}

Again the results remain well within the allowed range from 0.0003 to 0.0007. Therefore, we believe that the uncertainty quantification here is justified.

2. It seems like the truncation of the sum over intermediate states ought to be a significant source of uncertainty in this calculation. Here, a truncation to intermediate pions is considered; kaons are then added in to improve the calculation. It is not clear exactly how one would know a priori that including kaons is important, or how one would know that *only* kaons are important. Is there some structure in how one knows to add new contributions, or more importantly, some quantitative way to estimate the size of such truncation errors? This seems necessary to establish full control over uncertainties for a proper precision determination.

Response:

We agree with the referees that the truncation of the sum over intermediate states ought to be a source of uncertainty in this calculation. In dispersive analyses, contributions from the low-lying intermediates states are dominant since the dispersive integrals are constructed to be convergent by using subtractions. High-energy contributions are estimated with effective poles as well as parametrized into the subtraction polynomials.

Therefore, in this study, we have carefully accounted for the contributions of the lowest-lying hadronic channels $\pi\pi$ and $K\bar{K}$; the latter is important because the $f_0(980)$ scalar resonance couples strongly to the $K\bar{K}$ and also to $\pi\pi$. As discussed in the dispersive analysis of the nucleon σ -term form factor (see Ref. [72] cited in the main text), which shares a spectral function analogous to the trace GFF, the $\pi\pi$ - $K\bar{K}$ two-channel approximation should work well at low energies before inelasticities from the four-pion channels become important at about 1.3 GeV. Contributions from the four-pion intermediate states are approximated by the effective pole in our study. Such an approximation can indeed lead to theoretical uncertainties, which have been listed in the manuscript. As seen from Table I, because two subtractions have been employed in our Roy-Steiner equation analysis (as in Ref. [72]), the estimated uncertainty by varying the effective pole mass is not large. Therefore, we believe that the uncertainty quantification here is reasonable, which can be further improved once more data are available.

However, to be more conservative, we have decided to change “Precision” to the more narrative “Dispersive” in the title. Additionally, we have included a brief note in the main text to elaborate on this point.

3. The dispersive analysis must inherit *all* of the uncertainties of its inputs. This includes whatever experimental or lattice systematics, as well as truncation uncertainties in chiral perturbation theory. It doesn't seem like ChPT truncation uncertainties are taken into account, at least.

Response:

We completely agree with the referees' statement. In our paper, ChPT results are primarily utilized in the context of meson GFFs, more specifically, the NLO ChPT predictions for the slopes of the pion and kaon GFFs at $t = 0$ (Eqs. (45,46) in the Supplementary Material), i.e.,

$$\begin{aligned}\dot{\Theta}^\pi(0) &= 1 - 4 [L_{12}^r + 6 (L_{11}^r - L_{13}^r)] \frac{m_\pi^2}{F_\pi^2} - \frac{3}{2} \frac{m_\pi^2}{F_\pi^2} I_\pi + \frac{m_\pi^2}{2F_\pi^2} I_\eta = 0.98(2), \\ \dot{\Theta}^K(0) &= 1 - 4 [L_{12}^r + 6 (L_{11}^r - L_{13}^r)] \frac{m_K^2}{F_\pi^2} - \frac{m_K^2}{F_\pi^2} I_\eta = 0.94(14).\end{aligned}$$

Without having computed these quantities explicitly at the next-to-next-to-leading order (NNLO), the power of effective field theory allows us to estimate the truncation uncertainty to be of order $\mathcal{O}(m_\pi^4/\Lambda_\chi^4) = \mathcal{O}(2 \times 10^{-4})$ for the pion case, with $\Lambda_\chi = 4\pi F_\pi$. This is much smaller than the 2% uncertainty in the above quoted value of $\dot{\Theta}^\pi(0)$. In the kaon case, the NNLO contribution is of order $\mathcal{O}(m_K^4/\Lambda_\chi^4) = \mathcal{O}(3\%)$, again much smaller than the 15% uncertainty in the above quoted value of $\dot{\Theta}^K(0)$ —combining the two errors in quadrature leads to a negligible change. We have added a corresponding discussion in the main text to explain this point.

Note that applying such estimates to the NLO contribution would lead to truncation errors of order $\mathcal{O}(m_\pi^2/\Lambda_\chi^2) = \mathcal{O}(1.5\%)$ for the pion case and $\mathcal{O}(m_K^2/\Lambda_\chi^2) = \mathcal{O}(18\%)$ for the kaon case if we were to use only the LO results (that is, 1 for both cases). The above numerical values (2% and 6% corrections to the LO results for the pion and kaon cases, respectively) are indeed in line with such estimates.

4. There is one remaining precision comparison with lattice results, regarding $D(0)$, “it agrees with recent LQCD results at $m_\pi = 170$ MeV [55], and has a considerably smaller uncertainty.” We think this should be adjusted as with the other precision comparisons, as per our last comments about such comparisons not being well-posed.

Response:

We acknowledge the referee for raising this issue. As discussed above, we believe that our error estimates are reasonable. Nevertheless, we have followed the referee's suggestion to remove the statement “it agrees with recent LQCD results at $m_\pi = 170$ MeV [55], and has a considerably smaller uncertainty.”

5. It seems like there are some inputs which may *only* be obtained from lattice calculations, at least with the precision required to enable a high-precision dispersive determination. Is this actually the case, or are there alternatives?

Response:

The inputs to our dispersive analysis fall into three categories. The first part consists of ChPT predictions for the meson GFFs, specifically the slopes of the mesonic GFFs at $t = 0$. The second component is based on the S -wave and D -wave phase shifts, which are entirely determined by rigorous, data-driven dispersive analyses using the Roy(-like) equations in the literature. Additionally, for the nucleon GFFs, we require the partial wave amplitudes for $\pi\pi/K\bar{K} \rightarrow N\bar{N}$, also derived from data-driven dispersive analyses using the Roy-Steiner equation.

If high-precision lattice results for the meson GFFs become available in the future, the LECs in NLO ChPT can be determined, eliminating the need to estimate with the meson dominance model, and thus reduce that part of the uncertainties. This point has been mentioned above Eq. (18) of the main text. There, we also mention that the $\pi\pi$ scattering phase shifts up to 1.8 GeV from the very recent analysis in Ref. [93] can be used to improve the $\pi\pi$ - $K\bar{K}$ dispersive treatment beyond ~ 1.4 GeV, which will reduce the error due to the effective pole; such an analysis remains to be done.

Furthermore, since the GFFs can be connected to integrations of generalized parton distributions, weighted by the parton longitudinal momentum fraction, the precision of the GFFs may also be improved once high precision results for both quark and gluon GPDs are available.

We note that most of our remaining complaints are due to the interacting issues of 1) whether full quantification of uncertainties has been achieved in this work, and 2) the framing of these results as *already* precision determinations which may be fairly compared with others.

On 1), per our observations above, we don't think this has been achieved yet and might amount to substantial additional work. On 2), there are presently *no* determinations of the GFFs from any source with a comprehensive uncertainty budget, so there is not necessarily any sense in which a meaningful precision comparison could be achieved in the first place. Rather than inducing potentially unnecessary work, we want to note that essentially all of our concerns could be alleviated by some moderate reframings.

On 1), whether or not this satisfies the criteria of a proper precision determination, this determination of (especially the nucleon) GFFs is technically challenging, a significant advance over the previous state of the art, and at least *lays the groundwork* for precision dispersive determinations. This by itself is more sufficient to merit publication in Nature Communications. On 2), rather than framing this method as in competition with lattice methods (given that lattice inputs are required), a better comparison may be e.g. global fits for PDFs, GPDs, etc., which have recently been exploring including lattice inputs along with experimental data to improve precision. Reworking the paper in these terms and adding notes of any potentially large outstanding sources of uncertainty (see questions above) would address the essence of our concerns.

Response:

We are grateful to the referees for highly appreciating our work.

We understand the referees' concerns regarding uncertainty quantification and the issue of a precise comparison. We understand that it is challenging to reach a very high precision for all the GFFs (say, to reach a percent level uncertainty for the nucleon D -term). Yet, from the error budget in Table I of the main text, we believe that a percent level precision for the various radii listed therein has been achieved. However, to be more conservative, we have changed "Precision" to the more narrative "Dispersive" in the title. In the main text, we have removed statements concerning precision comparisons with lattice results. Additionally, we have added some notes in the main text to discuss why the uncertainty estimates are reasonable, explicitly addressing the concerns raised by the referees. A comparison with results from GPDs would also be challenging as the gluonic part could be difficult to extract from experimental measurements. In this sense, the dispersive approach is an invaluable tool for studying GFFs.

A comparison with various model results and model-dependent extractions from the J/ψ photo-production data has been included in the figures in the main text. In this further revised version, we have added one sentence on p.6, "*Given the substantial challenges of direct measurements of GFFs, especially their gluonic components, the dispersive determinations provide invaluable insights into nucleon structure.*" for some moderate reframings as suggested by the referees.

We hope that future combined efforts, including global fits for GPDs, lattice QCD calculations, and data-driven dispersive methods, will contribute to reaching a very high precision for all GFFs.

We would like to thank the referees again for their careful reading and insightful comments and suggestions. We hope that, with the above clarifications and revisions in the manuscript, the new version is suitable for publication in Nature Communications in its current form.

Second Report on "precise determination of nucleon gravitational form factors" by X.H. Gao et al.

The authors have done extensive revisions on the first version submitted, included a table of uncertainties and a Supplementary Materials which is very helpful in understanding some of technical details. Moreover, there are substantial changes in the text which reflected comments from previous round of reports. I am generally happy now with the paper, and recommend its publication with some optional revisions which can take into account my additional comments and feedback below. I don't need to see the revised version again, which shall be directly go to publication.

Here are some further comments:

1. Mass radius cannot be defined from the trace of EMT. This is wrong, because trace of EMT is not a mass operator as was emphasized in my first report. In all models of the nucleon, the mass is calculated from Hamiltonian as energy, same in lattice QCD calculations where two-point correlator at large Euclidean time is related to the energy of the system. In covariant perturbation theory, calculations of mass are identified by poles of Green's functions. This pole is invariant but when the momentum of the particle is zero, it is the rest energy. Masses must be ultimately related to scalars of a theory, like trace of EMT, or Higgs vev, but they are NOT mass directly. Therefore, there is only one mass radius which is the energy radius (Einstein said, "mass is energy"). If you prefer calling it energy radius, this is fine, but mass radius definitely sounds better because gravity is generally considered as from mass in non-relativistic limit.

2. My comment about the form factors of different operators with the same quantum numbers was not understood correctly. So let me ask in a different way. I can construct another second-rank tensor in QCD or meson theory. For example, I can multiply the EMT operator with scalar gluon fields FF to get another operator with the same quantum number. Surely, its form factor as a function of t will be very different. But your method seems to suggest they will have the same form factors (of course, the new operator is not conserved, but one can correct this by adding some additional term), If so, how does one know your result on t dependence is for the form factors of the very EMT? This question will not impact the result of your paper, but is interesting to consider.

3. I think the reply about non-uniqueness of EMT is not entirely correct. Your result might be the one that couples to gravity, but there are other possible forms of EMT which will have different form factors. There is no proof that the superpotential term vanishes in a physical eigenstate. This is related to the comment above. This question will not impact the result of your paper, but is interesting to consider.

4. I am happy with the reply about the previous literature on pion EMT form factors.

5. If looking at the trace anomaly operator, which is FF in chiral limit, I would guess scalar and tensor glueballs will make very significant contributions in the spectral function. I would not know how to quantify errors on these unknowns. I will allow to use the precision in title, but I can point out still other places one does not have control. One such place is Eq. (35) in the Supplement: without additional argument, simply take a linear form is a type of model for polynomials (maybe asymptotic large t analysis will help here).

6. The physics of a large scalar radius is clearly seen in MIT bag model where bag radius (or confinement radius) is related to this, and the bag constant is the QCD trace anomaly. The relevant physics has been commented in Ref. 104, and also in <https://arxiv.org/abs/2105.03974>. But I don't think ref. 107 and 108 are relevant.